# Concerted transformation of a hyper-paused transcription complex and its reinforcing protein

Philipp K. Zuber[1,9], Nelly Said [2], Tarek Hilal [2,3], Bing Wang [4], Bernhard Loll [2], Jorge González-Higueras[5,6], César A. Ramírez-Sarmiento [5,6], Georgiy A. Belogurov [7], Irina Artsimovitch [4] ✉, Markus C. Wahl [2,8] ✉ & Stefan H. Knauer [1,10] ✉

RfaH, a paralog of the universally conserved NusG, binds to RNA polymerases (RNAP) and ribosomes to activate expression of virulence genes. In free, autoinhibited RfaH, an α-helical KOW domain sequesters the RNAP-binding site. Upon recruitment to RNAP paused at an *ops* site, KOW is released and refolds into a β-barrel, which binds the ribosome. Here, we report structures of *ops*-paused transcription elongation complexes alone and bound to the autoinhibited and activated RfaH, which reveal swiveled, pre-translocated pause states stabilized by an *ops* hairpin in the non-template DNA. Auto-inhibited RfaH binds and twists the *ops* hairpin, expanding the RNA:DNA hybrid to 11 base pairs and triggering the KOW release. Once activated, RfaH hyper-stabilizes the pause, which thus requires anti-backtracking factors for escape. Our results suggest that the entire RfaH cycle is solely determined by the *ops* and RfaH sequences and provide insights into mechanisms of recruitment and metamorphosis of NusG homologs across all life.

In every cell, RNA synthesis is modulated by accessory proteins that bind to RNA polymerase (RNAP) and nucleic acids and adjust gene expression to cellular demands. Among these factors, NusG stands out as the only regulator conserved across all domains of life[1]. NusG proteins bind to RNAP genome-wide[2,3] to promote efficient synthesis and folding of the nascent RNA[4–7] and consist of a NusG N-terminal (NGN) domain flexibly connected to one C-terminal Kyprides, Ouzounis, Woese (KOW) domain (or several in eukaryotes). The NGNs share α/β topology, bind to a conserved site on RNAP, and are sufficient for direct effects on RNA synthesis[5,8–13]. The β-barrel KOW domains

contact diverse proteins to couple transcription to RNA folding, modification, splicing, nucleosome remodeling, translation, and other cellular processes[6,14–21].

Many cellular genomes encode specialized paralogs of NusG[22]; in bacteria, they are required for conjugation and biosynthesis of antibiotics, capsules, lipopolysaccharides, and toxins, and are thus vital for fitness, pathogenesis, and evolution[23]. NusG paralogs function alongside NusG to control the expression of just a few genes, which are essential only under some conditions, e.g., during infection[24]. The regulatory logic that underpins this division of labor is well understood

[1]Biochemistry IV-Biophysical Chemistry, Universität Bayreuth, Bayreuth, Germany. [2]Institute of Chemistry and Biochemistry, Laboratory of Structural Biochemistry, Freie Universität Berlin, Berlin, Germany. [3]Research Center of Electron Microscopy and Core Facility BioSupraMol, Freie Universität Berlin, Berlin, Germany. [4]Department of Microbiology and Center for RNA Biology, The Ohio State University, Columbus, OH, USA. [5]Institute for Biological and Medical Engineering, Schools of Engineering, Medicine and Biological Sciences, Pontificia Universidad Católica de Chile, Santiago, Chile. [6]ANID, Millennium Science Initiative Program, Millennium Institute for Integrative Biology, Santiago, Chile. [7]Department of Life Technologies, University of Turku, Turku, Finland. [8]Macromolecular Crystallography, Helmholtz-Zentrum Berlin für Materialien und Energie, Berlin, Germany. [9]Present address: MRC Laboratory of Molecular Biology, Cambridge Biomedical Campus, Cambridge, UK. [10]Present address: Bristol-Myers Squibb GmbH & Co. KGaA, Munich, Germany.
✉e-mail: artsimovitch.1@osu.edu; markus.wahl@fu-berlin.de; Stefan.knauer@bms.com

in *Escherichia coli*. NusG dynamically interacts with almost every transcribing RNAP[3] and determines the fate of the nascent RNA by either suppressing or promoting Rho-dependent termination. On translated mRNAs, NusG can bridge RNAP and ribosome[20,21] whereas on rRNA, NusG is part of an antitermination complex[6]; in both cases, the nascent RNA and RNAP are shielded from Rho. On antisense, aberrant, and xenogeneic RNAs, NusG KOW binds to Rho to induce premature termination[25–27].

Conversely, the expression of several xenogeneic operons critically depends on NusG paralog RfaH[28]. RfaH, but not NusG, associates with RNAP transcribing these genes[29], even though NusG vastly outnumbers RfaH in the cell[30]. A combination of sequence-specific recruitment and fold-switching-controlled autoinhibition ensures that RfaH finds its targets while not compromising the essential function of NusG[23]. In free RfaH, the RNAP-binding site on NGN is masked by its KOW domain adopting an α-helical hairpin (KOW$^\alpha$)[12]. RfaH recruitment requires a 12-nucleotide (nt) operon polarity suppressor (*ops*) sequence (Fig. 1)[31], which induces RNAP pausing[32]. Upon binding to the *ops*-paused elongation complex (*ops*PEC), RfaH is activated through domain dissociation[33]. The released NGN is accommodated on RNAP and converts the enzyme into a pause-resistant state[13], whereas the freed KOW refolds into a NusG-like five-stranded β-barrel (KOW$^\beta$) and binds to ribosomal protein S10[33] to couple transcription to translation[34]. Notably, RfaH recruitment to RNAP and ribosome must be tightly orchestrated: *ops* is the only chance for RfaH to load onto transcribing RNAP, and the ribosome must be captured by RfaH between *ops* and the start codon, located within 100 nts downstream. Failure of either recruitment reduces gene expression up to several hundred folds[28,34].

Understanding such synchronicity requires elucidation of minute structural detail. While much insight has been provided by structures of autoinhibited and activated RfaH[12,13,33,35], their limitations leave several key questions unanswered. First, what features of *ops*PEC render it exceptionally efficient at recruiting RfaH, which is present at fewer than 100 copies per cell? Second, in autoinhibited RfaH, the RNAP-binding site is partially occluded—how does RfaH bind to *ops*PEC? Third, how is RfaH domain dissociation triggered upon binding to *ops*PEC? Fourth, after accommodation of RfaH, how are ribosome recruitment and pause escape achieved? In this work, we sought to answer these questions by comparing structures of *ops*PECs alone and bound to the autoinhibited and activated RfaH. We show that the formation of an *ops* hairpin (*ops*HP) in the non-template (NT) DNA strand stabilizes a swiveled, pre-translocated paused state with an extended bubble and a 10-base pair (bp) RNA:DNA hybrid. Using the *ops*HP as a handle, autoinhibited RfaH docks onto *ops*PEC near its final binding site, forming a transient encounter complex where the KOW domain is primed for activation. At the same time, RfaH twists the *ops*HP, further expanding the hybrid to 11 bp and thus hyper-stabilizing a pause. This state persists even after full RfaH activation and accommodation and requires accessory factors for escape. Finally, our molecular dynamics simulations are consistent with previous evidence that the KOW α-to-β fold-switch spontaneously occurs after the initial domain separation, with the RNAP-RfaH contacts being dispensable for the fold-switch. Our results portray a remarkably economical mechanism of deoxyriboregulation of RNAP, in which a short 12-nt region of NT DNA directs major conformational changes in the transcription machinery that trigger further modulation via a metamorphic accessory factor, ultimately supporting the synthesis of vital proteins.

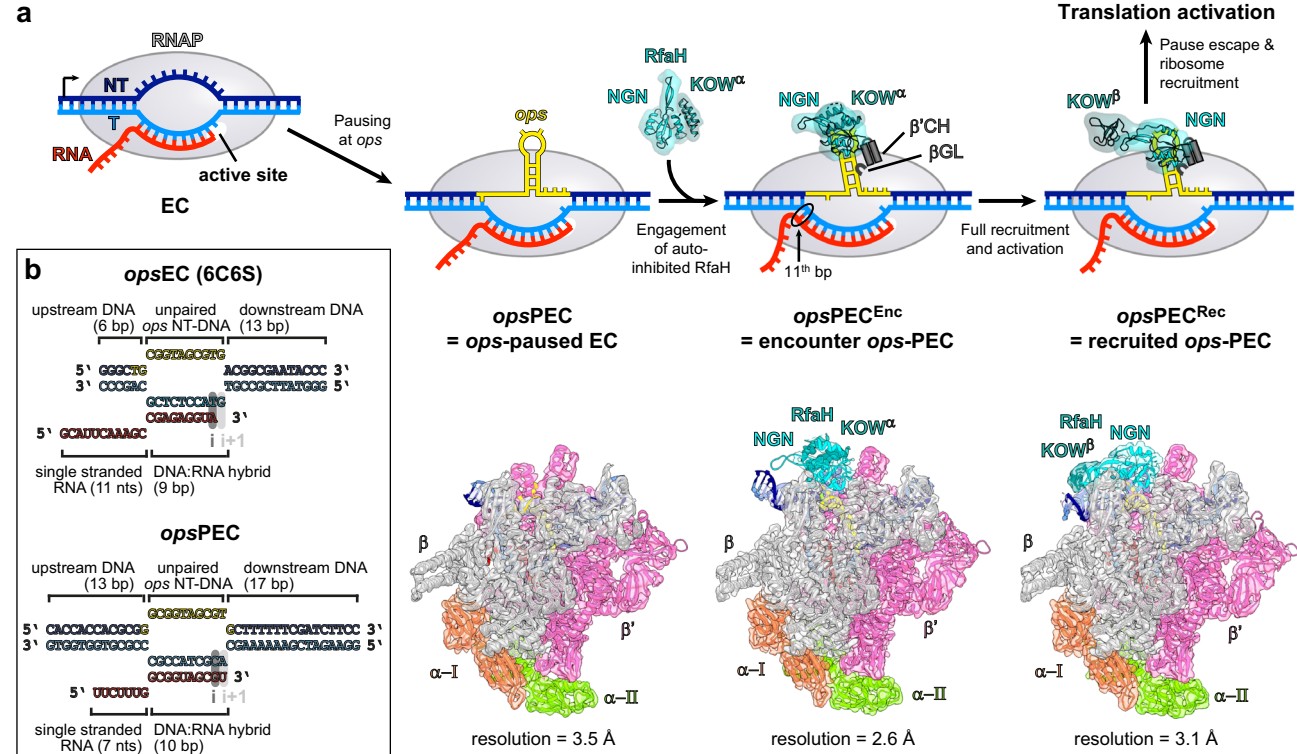

**Fig. 1 | Steps in RfaH recruitment and activation. a** Mechanisms of RNAP pausing at *ops* and RfaH recruitment revealed by cryoEM structures. In *ops*PEC, the NT-DNA strand folds into a hairpin that recruits autoinhibited RfaH to form the PEC$^{Enc}$. During subsequent RfaH activation, KOW$^\alpha$ is released from NGN, which establishes stable contacts with RNAP in the resulting *ops*PEC$^{Rec}$, while KOW$^\alpha$ refolds into KOW$^\beta$ to set up a stage for recruitment of a ribosome to initiate translation. The cryoEM densities (transparent surfaces) and accompanying models (cartoons) of *ops*PEC, *ops*PEC$^{Enc}$, and *ops*PEC$^{Rec}$ are shown below. **b** Nucleic acids scaffolds used for the assembly of a post-translocated *ops*EC[13] (top) and the pre-translocated *ops*PEC used in this study (bottom). In this and other figures, the NT-DNA is shown in dark blue, the T-DNA in light blue, the *ops* element in yellow, and RNA in red.

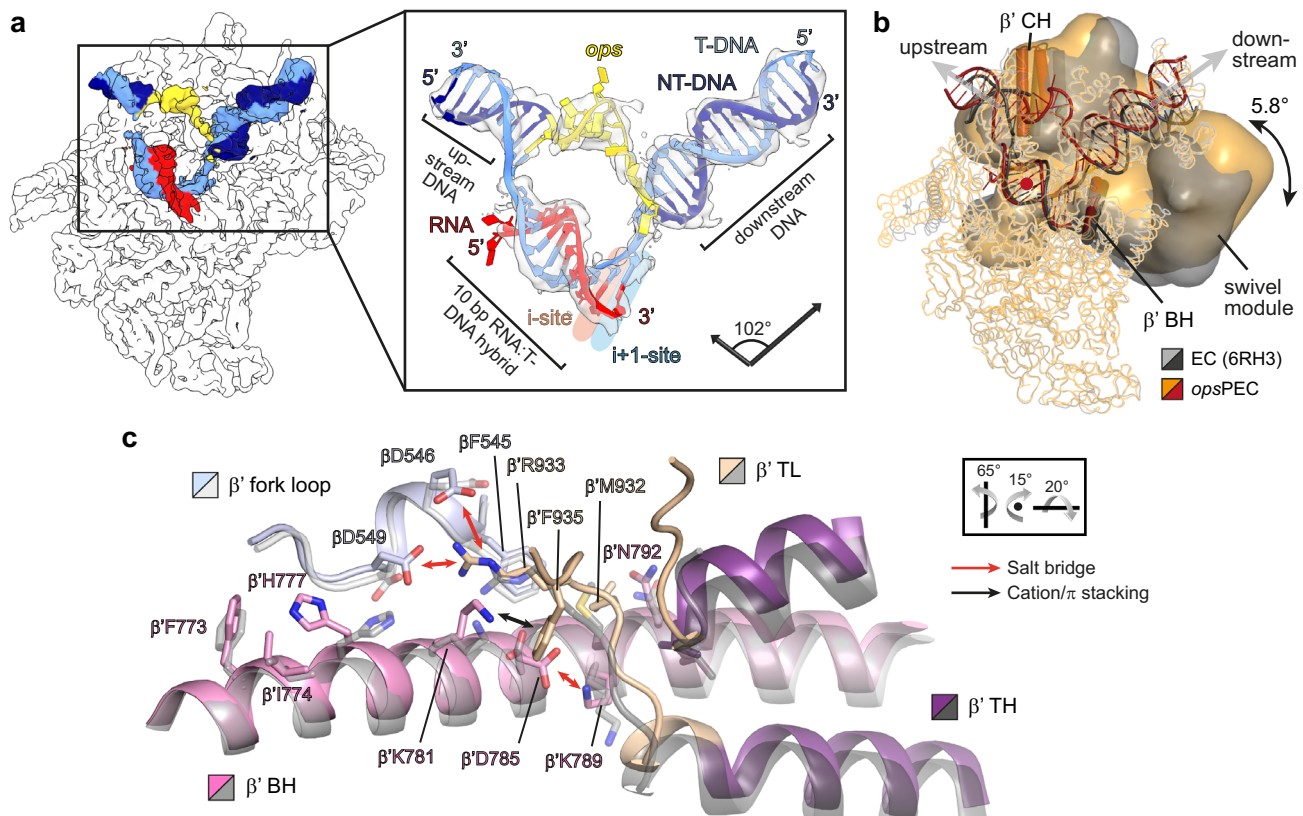

**Fig. 2 | Structural basis of transcriptional pausing in the *ops*PEC. a** Left: CryoEM map of RNAP is shown as a white outline, nucleic acids maps—as solid surface. Right: Model of the nucleic acid scaffold (cartoon) with its associated cryoEM density (transparent surface). In this and other figures, the arrows represent the helix vectors of the up- and downstream DNA duplexes, and the angle between them is given. **b** A structural overlay of a pre-translocated EC (PDB ID 6RH3) and *ops*PEC superimposed on the core module (mainly α- and β-subunits; shown as ribbons); the swivel module is represented as Gaussian surface. The swivelling angle and the approximate rotation axis (red dot) are shown. The β'BH and β'CH are displayed as cartoon tubes to illustrate the orientation of the swivelling axis (parallel to β'BH). **c** Superposition of β'BH and β'TL of the *ops*PEC and a post-translocated EC (PDB ID 6ALF; gray). Arrows indicate interactions within residues (sticks) that trap the inactive β'TL; orientation relative to panel a is indicated.

## Results

### RNAP pauses at the *ops* element in a pre-translocated state

To enable RfaH recruitment, RNAP (a five-subunit α₂ββ'ω complex in *E. coli*) pauses at the *ops* site, yielding a pre-translocated PEC[36] (Fig. 1a). A previous analysis revealed the structure of RfaH bound to a non-paused EC (RfaH-*ops*EC)[13] assembled on a scaffold (Fig. 1b; Supplementary Figs. 1, 2a) with the NT DNA *ops* element, a template (T) DNA strand that lacked complementarity in the last 10 *ops* nucleotides, most notably the NT 3'-terminal G12 (thereafter, all nucleotides are numbered to reflect their positions in the *ops* element, with RNA and T-DNA nts denoted with an R/T superscript), and a 9-bp hybrid, thereby favoring the post-translocated state. Finally, the scaffold had a 6-bp upstream DNA duplex, possibly precluding nucleic acid repositioning via RNAP or factor contacts to more distal DNA regions. Thus, although this structure captured the details of RfaH interactions with RNAP and DNA, the molecular basis of initial *ops*-mediated pausing is presently unknown.

Here, we assembled an *ops*PEC on a fully complementary (c) scaffold harboring the *ops* site, an extended upstream duplex (up to 14 bps), and an RNA that could form a hybrid of up to 11 bps (Fig. 1b) and elucidated its structure by cryogenic electron microscopy (cryoEM) in combination with single-particle analysis (SPA; Supplementary Figs. 1, 2; Supplementary Table 1). The particles were highly homogeneous and in the cryoEM reconstruction, almost all regions of RNAP, all DNA nts and 10 nts of RNA were well resolved, with fragmented density for RNA outside of the exit channel.

A PEC formed at *ops* is biochemically distinct from a PEC stabilized by an RNA hairpin, e.g., *his*PEC[36]. Consistently, while in structures of *his*PECs the RNA:DNA hybrid adopted a half-translocated state[13,37] (RNA post-translocated, DNA pre-translocated), our *ops*PEC resides in the pre-translocated state (Fig. 2a). RNAP is paused at U11, and no unpaired T-DNA nucleotide is present in the i + 1 position to receive an incoming rNTP. Compared to an elongation-competent, post-translocated EC[38], *his*PECs adopt a swiveled state, in which a swivel module (clamp, dock, shelf, SI3, and a C-terminal segment of the β' subunit) is rotated by ~3° about an axis perpendicular to the plane defined by the axes of the upstream DNA duplex and the RNA:DNA hybrid[13,37]. Swiveling is thought to stabilize the paused state by counteracting folding of the catalytic β' trigger loop (TL)[13,37]. Consistent with the exceptionally strong pausing at *ops*[39], *ops*PEC undergoes particularly pronounced swiveling of 5.8° (Fig. 2b); TL is unfolded (Fig. 2c) and the β' SI3 domain is in the open conformation. In *ops*PEC, but not in EC[38], the TL β'R933 forms salt bridges with βE546 and βD549, an interaction that may stabilize the unfolded TL (Fig. 2c).

In contrast to all other structures of factor-free ECs, the NT strand is fully defined in the cryoEM reconstruction (Fig. 2a). Bases G2-C9 form a hairpin that rests on top of the β' rudder R314 (Fig. 3a, b). The *ops*HP stem comprises two Watson-Crick (WC) bps (G2:C9 and C3:G8) and a Saenger XI bp (G4:A7) and is stabilized by positively charged residues of the β lobe (R201, R371, R394), β protrusion (R470, R473), and β' rudder (K321); G1 forms the most proximal bp of the upstream DNA duplex (Fig. 3b). Accommodation of the *ops*HP at the β lobe/

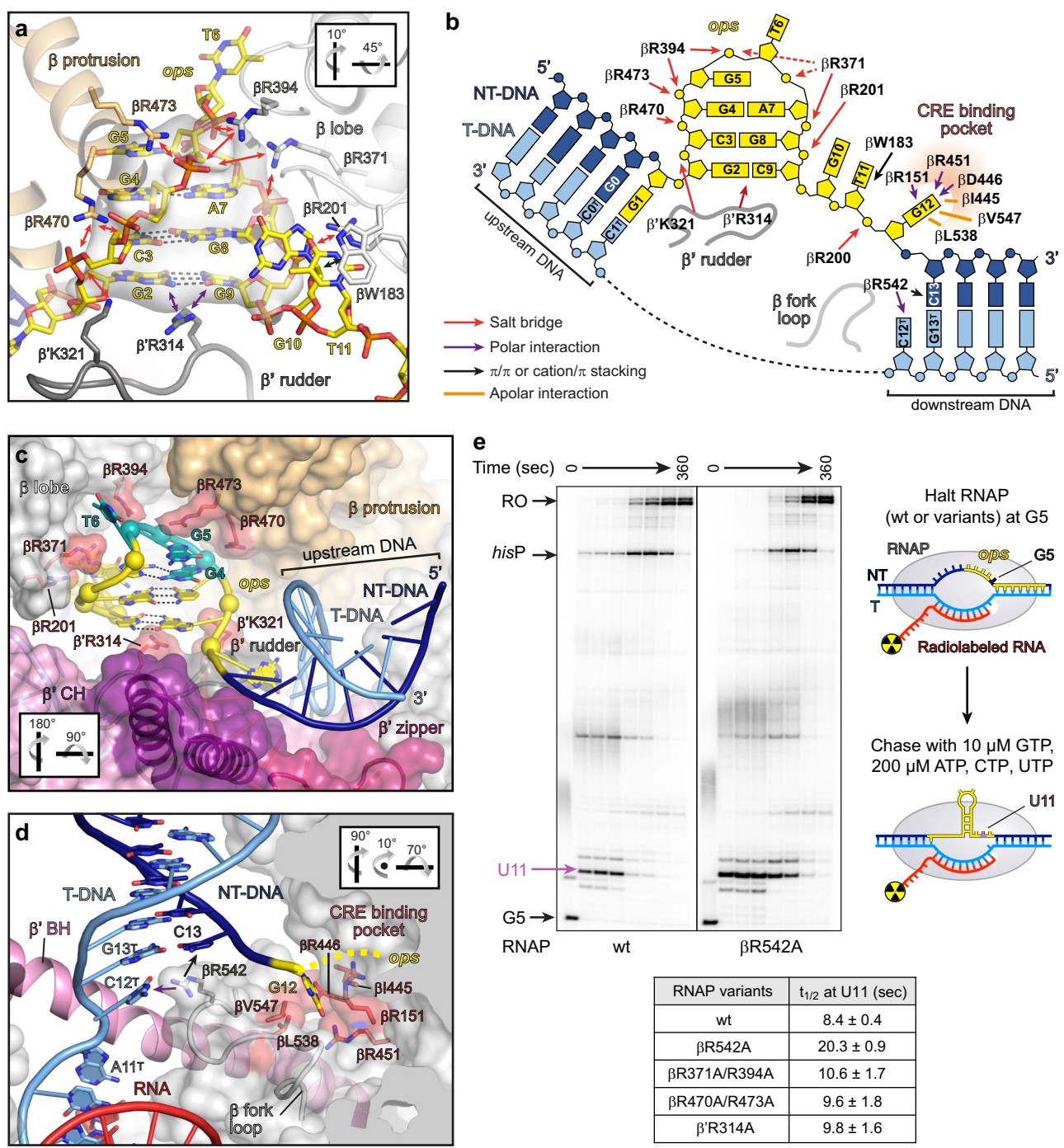

**Fig. 3 | RNAP/nucleic acid interactions in the opsPEC. a** The *ops* DNA hairpin in the NT strand. The *ops*HP and side chains of interacting residues are shown as sticks, the RNAP elements—as transparent cartoons. The cryoEM density is shown as gray surface, H-bonds formed by the *ops* bps—as dashed lines. **b** The RNAP:*ops* interactions. DNA nucleotides are depicted as blocks (bases), rings (ribose), or circles (phosphates), respectively. Salt bridges, polar or apolar interactions with selected RNAP residues are color-coded and indicated by arrows or lines. **c** Stabilization and accessibility of the *ops*HP within the main channel cleft. RNAP is shown as transparent surface; selected structural elements—as colored cartoons. Side chains of *ops*-interacting residues are in salmon, the *ops* nts interacting with RfaH—in mint. **d** Separation of the *ops* G12:C12$^T$ bp at the downstream DNA fork junction. RNAP (sliced at the active site; β′ is omitted for clarity) is shown as

transparent surface, nucleic acids as cartoon/sticks. G12 is inserted into the CRE pocket (red), whereas C12$^T$ is stabilized by β fork loop 2; red arrows indicate interactions with βR542. Orientations in (**a**), (**b**), and (**d**) are relative to the standard view (Fig. 2a). **e** Effects of RNAP residue substitutions on pausing. Halted radiolabeled G5 ECs were formed with wt or mutationally-altered RNAPs. Single-round elongation assays were carried out as described in Methods. Samples withdrawn at 0, 10, 20, 30, 60, 90, 180, and 360 s were analyzed on a urea-acrylamide gel; a representative gel is shown. The positions of *ops* G5 and U11, *his*P, and run-off (RO) RNAs are indicated with arrows. For each RNAP variant, the assay was repeated three times with similar results; the half-life ($t_{1/2}$) of pausing at U11 is presented as mean ± SD. Source data are provided as a Source Data file.

protrusion pushes the upstream DNA away from the β protrusion and against the β′ zipper and clamp helices (CH), promoting swiveling (Figs. 2b, 3c). Consequently, the upstream and downstream DNA duplexes span an angle of 102°, vs ~129° in the canonical EC (Fig. 2a and Supplementary Fig. 2b).

The two-nt opsHP loop (G5-T6) is required for specific recognition by RfaH[13,35]. In opsPEC, this loop is located between the upstream DNA channel and the main channel of RNAP (Fig. 3c). The T6 base is completely flipped outwards and is highly flexible (Figs. 2a and 3b) and G5 is also rotated outwards. Immediately downstream of the opsHP, the NT DNA changes direction; G10 and T11 move away from the transcription bubble and form an extended stack with βW183 that is laterally stabilized by βD199 and βR200 (β lobe; Fig. 3a, b). G12 is embedded in the core recognition element (CRE) pocket[40] and is unable to pair with C12$^T$; instead, the following C13 is diverted to form the first bp of the downstream DNA duplex (Fig. 3d).

In the pre-translocated EC, 10 DNA bp are melted to form a 10-bp RNA:DNA hybrid[41]. Formation of the opsHP requires 11 single-stranded (ss) NT nts−thus, an additional upstream DNA bp is melted and the last unpaired T-strand nt, C12$^T$, remains stacked on the downstream duplex and does not move into the templating i+1 position (Fig. 3d). Consequently, the bubble is compressed as compared to ECs, causing a sharper angle between the upstream and downstream duplexes.

The side chain of βR542 occupies the position of G12, engaging the WC face of C12$^T$ (Fig. 3d). This interaction may stabilize the pre-translocated state or promote local melting of downstream DNA to facilitate pause escape. We substituted βR542 for an alanine and characterized the pausing behavior of wild-type (wt) and β$^{R542A}$ RNAPs in vitro on a template that contains a strong T7A1 promoter followed by the ops and his pause elements[12]. On this template, RNAP can be halted at ops G5 in the absence of UTP; thus synchronized, α$^{32}$P-labeled G5 ECs are restarted upon the addition of all NTPs. RNAP pauses at the ops (U11) and his pause sites before making the run-off (RO) RNA. The wt RNAP paused at U11 with a half-life of 8 seconds, whereas βR542A substitution delayed escape ~2.5 fold (Fig. 3e), suggesting that βR542 promotes escape from ops, in contrast to its effect at the elemental pause[42]. The βR542 sidechain can interact with the edge of the downstream DNA (e.g., PDB IDs 5VOI, 7YPA, 8FVW) or NT DNA −1/+1 nts (e.g., PDB IDs 8EG7, 8EG8, 8EH8), and βR542 effects on pause escape may thus differ depending on the sequence context. In opsPEC, βR542 presumably favors DNA separation by temporarily replacing base pairing with the protein-T DNA interaction.

To evaluate the role of residues that appear to stabilize the opsHP, we substituted arginine residues in the β lobe (R371A/R394A), β protrusion (R470A/R473A), or β′ rudder (R314A) and characterized pausing of the RNAP variants in vitro. We found that neither substitution had significant effects on pausing (Fig. 3e), as was also observed with base substitutions that destabilized the opsHP stem[35].

To assess the basis of the transcription bubble extension, we determined a 3.0 Å structure of an opsPEC assembled on a partially non-complementary (nc) scaffold (Supplementary Fig. 2c, e, g and Supplementary Table 2). The nc-opsPEC was virtually identical to opsPEC assembled on the complementary scaffold (Supplementary Fig. 2d–h). Thus, the extended ss regions in the opsPEC form independently of the precise sequences in the bubble and the hybrid, suggesting that this extension principally depends on the NT-DNA hairpin.

## RfaH recruitment proceeds via a hyper-paused encounter complex

In autoinhibited RfaH, the KOW$^α$ masks the β′CH-binding site on NGN[12], yet NMR data show that autoinhibited RfaH binds RNAP[33], suggesting the existence of a transient encounter complex, opsPEC$^{Enc}$. To image this complex, we used a F51C,S139C RfaH variant (RfaH$^{CC}$) locked in the autoinhibited state by a disulfide bridge between NGN and KOW$^α$

(Fig. 4a). We showed that RfaH$^{CC}$ is fully active under reducing conditions[12], and far-UV CD and 2D [$^1$H,$^{15}$N]-HSQC spectra confirmed expected RfaH$^{CC}$ properties (Supplementary Fig. 3).

We mixed RfaH$^{CC}$ with the opsPEC under non-reducing conditions and determined the structure of the ensuing opsPEC$^{Enc}$ at 2.6 Å resolution (Fig. 1a and Supplementary Table 1). Except for the β′ zinc binding domain (ZBD), most RNAP elements and the nucleic acid regions were well defined in the cryoEM reconstruction. In addition, there was a clear density for RfaH$^{CC}$. The opsPEC$^{Enc}$ is swiveled, albeit less strongly than opsPEC (4.9° vs. 5.8°; Supplementary Fig. 4a), with an unfolded TL and an open SI3. Correlating with reduced swiveling, the β′CH are slightly displaced from the β lobe/protrusion, and the angle between upstream and downstream DNA duplexes is increased compared to opsPEC (~116°) but remains smaller than in the canonical EC (Fig. 4b). Therefore, sufficient space exists between the β lobe/protrusion and β′CH for NGN to bind the opsHP and RNAP (Fig. 4b, c) but the KOW$^α$-NGN interaction prevents full accommodation of RfaH$^{CC}$ as observed in RfaH-opsEC[13].

In RfaH-opsEC[13], NGN is positioned across the RNAP main channel, helices α1 and α2 contact the β protrusion and β lobe, respectively, and the opposite open flank of the central β-sheet and α3 contact the β′CH. In opsPEC$^{Enc}$, α2 packs similarly, whereas NGN is rotated about the α2 axis towards the β protrusion (Fig. 4c and Supplementary Fig. 4b, d, e). Helix α1 is moved closer to, and interacts more intimately with, the β protrusion, while the loop preceding α3 is displaced from the β lobe and instead interacts with the β′ clamp region neighboring the β′CH. Helix α3 is thereby moved away from the β′CH, and its rough position is instead occupied by the α2* helix of KOW$^α$ (* denotes secondary structure elements in KOW; Fig. 4c, d and Supplementary Fig. 4b).

In opsPEC$^{Enc}$, the opsHP is engaged by NGN (Fig. 4e, f and Supplementary Fig. 4d) as observed in the crystal structure of the RfaH-ops binary complex[35]; i.e., the opsHP is bound at NGN opposite KOW$^α$, T6 is inserted into a positively charged NGN pocket, and G5 packs against a neighboring surface (Fig. 4e and Supplementary Fig. 4d). Nucleobase-specific interactions between NGN and the opsHP loop (Fig. 4f) show that RfaH reads out the NT-DNA sequence already in opsPEC$^{Enc}$. T6 also stacks on βY62 (Supplementary Fig. 4c), an interaction not observed in RfaH-opsEC[13]. The same positively charged RNAP residues as in opsPEC contact the opsHP stem, but at different chemical functionalities or nts. As a result, T6 is snuggly sandwiched between NGN and RNAP, and the entire opsHP is tightly restrained, as indicated by its well-defined density (Fig. 4b, f).

Upon grabbing a hold of its loop, RfaH$^{CC}$ twists the opsHP and redirects its tip towards the upstream DNA, while G10 unstacks from T11, which retains stacking interactions with βW183 (Fig. 5a, b and Supplementary Fig. 4c). Furthermore, RfaH$^{CC}$ Y8 and R11 contact the sugar-phosphate backbone of upstream DNA on the major groove side (Fig. 4d); based on these DNA "anchors", the proximal T-DNA branch of the upstream duplex is pulled against the β′ rudder (Fig. 5b), which acts like a strand separator, displacing C1$^T$ from G1 to melt an additional upstream DNA bp (Fig. 5). A β-hairpin loop (HL; M32-L50) of NGN is well-defined, and R40 at the HL tip contacts the more distal upstream DNA backbone (Fig. 4d, e). In opsEC-RfaH, the HL is disordered, possibly due to the short upstream DNA duplex employed[13].

Upstream DNA melting leaves G1 unpaired, and its nucleobase stacks with the neighboring upstream nt (G0); the β′CH residues stabilize G1 (β′R271, β′N274) and the new most proximal upstream DNA bp (β′R270). Most importantly, the liberated C1$^T$ is paired with G1$^R$, extending the hybrid to 11 bps (Fig. 5). As in opsPEC, β′L255 and β′R259 (β′ lid) cap the upstream edge of the hybrid, leading to its further compression, an increase in diameter by ~1 Å, and, thus, a more A-like conformation compared to the 10-bp hybrid in opsPEC (Fig. 5b, c). A weak density for the most downstream bp (Fig. 4b) suggests that the pre-translocated hybrid is pushed backward, further counteracting

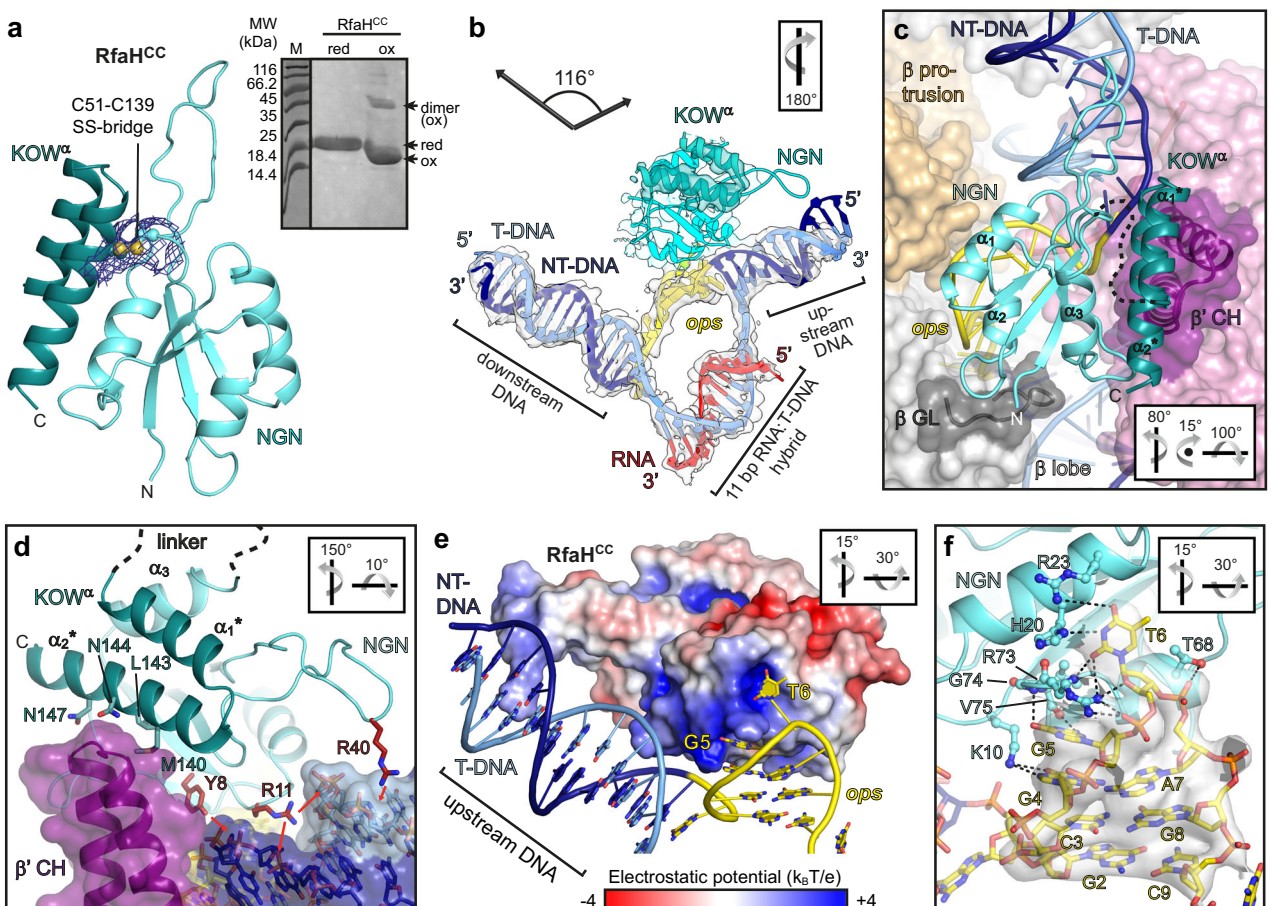

**Fig. 4 | RfaH recruitment proceeds via an encounter complex. a** C51-C139 bridge (balls and sticks, with cryoEM density as blue mesh) prevents domain separation in RfaH$^{CC}$. SDS-PAGE analysis of RfaH$^{CC}$ in reducing (red) or non-reducing (ox) conditions. Under oxidizing conditions, the intra-molecular bridging leads to faster migration, and intermolecular bridging gives rise to dimers. **b** RfaH$^{CC}$ bound to the nucleic acid scaffold shown as cartoon with the associated cryoEM densities (transparent surface). **c** RfaH$^{CC}$ (cyan/mint, with several α-helices labeled and the interdomain linker shown as a dashed line) is wedged between the β'CH and *ops*HP. RNAP is shown as transparent surface, with relevant elements as color-coded cartoons. **d** The β'CH act as a wedge to induce RfaH domain separation. In RfaH$^{CC}$ and β'CH, side chains of interacting residues are shown as sticks. Nucleic acids are represented as transparent surface/sticks. RfaH$^{CC}$ residues forming polar interactions (red arrows) with the upstream DNA are in salmon. **e** RfaH$^{CC}$ binds *ops* and upstream DNA via positively charged patches. The electrostatic potential of RfaH$^{CC}$ is mapped on its molecular surface. **f** Side chains of NGN residues contacting *ops* are depicted as sticks (salmon). The *ops* DNA is shown as sticks along with the cryoEM density of the *ops*HP (transparent surface). H-bonds and electrostatic interactions are indicated by dashed lines.

translocation. Thus, pausing not only persists but is reinforced during initial RfaH recruitment.

We wondered how strongly the initial docking of RfaH$^{CC}$ onto the *ops*HP drives upstream DNA melting and hybrid expansion. Therefore, we assembled an EC on a scaffold in which a WC bp in the upstream DNA would have to be disrupted and formation of a non-WC C:U pair would have to be "forced" into the 11-bp hybrid (Supplementary Fig. 5). Strikingly, the 3.1 Å structure revealed that the ensuing RfaH$^{CC}$-bound complex was virtually identical to *ops*PEC$^{Enc}$, with the 11-bp hybrid, even though the C:U bp does not energetically fully compensate for the lost DNA bp (Supplementary Fig. 5e–g). We conclude that docking of RfaH provides a strong driving force for hybrid expansion, leading to a hyper-paused *ops*PEC$^{Enc}$.

The overall conformation of RfaH$^{CC}$ closely resembles the structure of isolated RfaH[12], but embedding of RfaH$^{CC}$ between the β lobe, β protrusion and β'CH leads to a slight displacement of KOW$^α$ relative to NGN (Fig. 4c, d and Supplementary Fig. 4b, e). While full displacement of KOW$^α$ is prevented by the disulfide bridge, the β'CH tip acts like a wedge that starts to insert between NGN and KOW$^α$. Concomitantly, α1* and α2* are unwound by two N-terminal turns and one C-terminal turn, respectively (Supplementary Fig. 4e). Thus, upon initial docking to *ops*PEC, autoinhibited RfaH takes a handle of the *ops*HP loop to pull

its own KOW$^α$ against the β'CH, generating steric conflicts that prime KOW$^α$ dissociation and subsequent refolding.

## The hyper-paused state persists after full accommodation of RfaH

To follow the complete RfaH recruitment and activation, we determined cryoEM/SPA structures of *ops*PECs assembled on c- and nc-scaffolds with RfaH$^{wt}$ (Figs. 1, 6a, Supplementary Fig. 6 and Supplementary Tables 1 and 2). To facilitate possible conformational changes, we incubated the samples at 37 °C for 10 min before vitrification and imaging. CryoEM reconstructions followed by 3D variability analysis (3DVA) revealed two nearly identical states (Fig. 6a). We focus on a complex assembled on the c-scaffold in which both RfaH domains are visible (state 1 in Fig. 6a; thereafter designated as *ops*PEC$^{Rec}$) and discuss the alternative state below.

RfaH is activated (Fig. 6b), with NGN fully embedded between the β lobe, β protrusion and β'CH. As compared to *ops*PEC$^{Enc}$, NGN is rotated about the α2 axis, so that α1 is slightly displaced from the β protrusion, while α3 moves to contact the β'CH (Fig. 6c). The *ops*PEC$^{Rec}$ retains a swiveled conformation, albeit with a reduced angle of 3.3°. Thus, RfaH binding per se does not prevent swiveling as previously suggested[13].

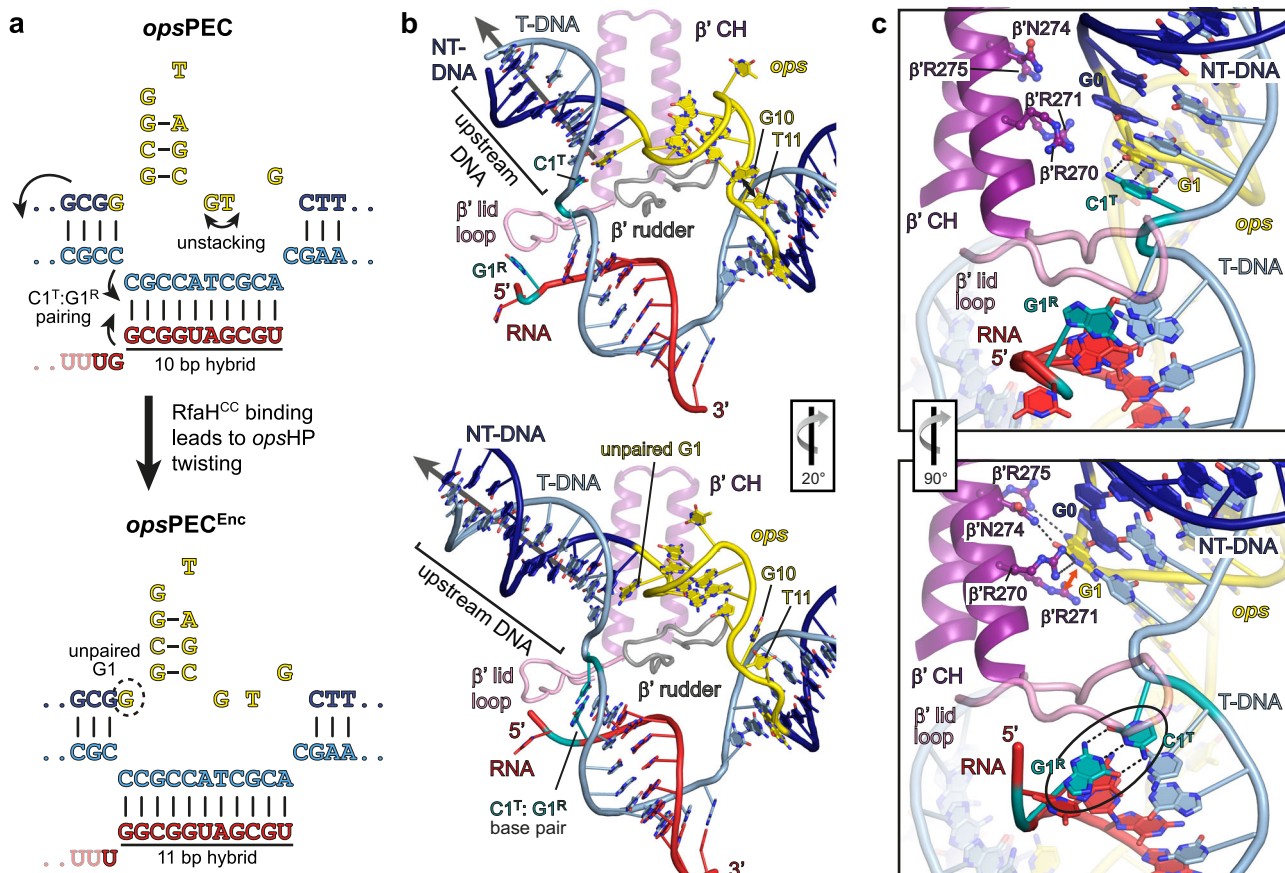

**Fig. 5 | Rearrangements of *ops*PEC (top) into the hyper-paused *ops*PEC$^{Enc}$ (bottom) following binding of RfaH$^{CC}$. a** A summary of nucleic acid changes. **b** Nucleic acids (cartoon/stick) and β' lid loop, β' rudder loop and β'CH (cartoon). C1$^T$ and G1$^R$ form an 11$^{th}$ bp in *ops*PEC$^{Enc}$ (cyan). The upstream DNA vectors are indicated by arrows. **c** A close-up views of the upstream fork junction. Selected β'CH side chains are shown as sticks; polar (dashed lines) and stacking (red arrow) interactions are indicated. Orientations are relative to the standard view (Fig. 2a).

The *ops*HP conformation and position are essentially unaltered compared to *ops*PEC$^{Enc}$ and very similar to RfaH-*ops*EC (Figs. 4b and 6b). However, the loop preceding α1 is moved closer to the upstream DNA and engages in more intimate interactions with the sugar-phosphate backbone on the major groove side; the HL-DNA contacts are also maintained, albeit to a more proximal region of the upstream duplex (Fig. 6c). We observed clear density for the refolded KOW$^β$, which is positioned on top of the HL and contacts the β'ZBD, which becomes ordered (Fig. 6b, c). KOW$^β$ E124 and F126 sandwich the β'ZBD K87, and the upstream DNA is displaced towards the β protrusion and flap (Fig. 6c, d). In RfaH-*ops*EC, KOW$^β$ was in a similar location[13], but also interacted with the β flap-tip helix (FTH). In *ops*PEC$^{Rec}$, in contrast, the βFTH remains disordered, possibly because the longer upstream DNA prevents close approach to KOW$^β$. Notably, the binding site for S10 is exposed in KOW$^β$.

The upstream duplex is pushed against the β' rudder, maintaining the additional melted bp, and the hybrid is compressed and pre-translocated in *ops*PEC$^{Rec}$ assembled on either the c- or nc-scaffold, indicating that the strong driving force for upstream DNA melting and hybrid expansion is maintained in *ops*PEC$^{Rec}$. In summary, *ops*PEC$^{Rec}$ remains hyper-paused, with KOW$^β$ poised to engage a ribosome to form an RfaH-bridged expressome.

**Refolding landscape of RfaH upon recruitment to *ops*PEC**
Ribosomal interactions with KOW$^β$ are critical for the cellular function of RfaH[34]. Thus, the KOW fold-switch is the final step in RfaH activation. Our structures capture the autoinhibited and activated states of RfaH-bound *ops*PECs but provide no information about their interconversion. Molecular dynamics (MD) simulations used to interrogate the KOW switch[43] were performed with the isolated RfaH or KOW. To explore the fold-switch of RfaH bound to *ops*PEC, we generated an all-atom dual-basin structure-based model (SBM), such as those employed to study massive structural transitions of influenza hemagglutinin[44]. In 500 independent runs performed using a dual-basin SBM created based on *ops*PEC$^{Enc}$ and *ops*PEC$^{Rec}$, KOW underwent a complete α-to-β fold-switch.

Figure 7 shows the refolding landscape of RfaH projected onto the fraction of interdomain (ID) contacts ($Q_{ID}$) and the difference in the fraction of native contacts formed with respect to either KOW$^α$ or KOW$^β$ ($Q_{diff}$). To visualize KOW$^β$ when $Q_{ID} = 0$, we employed the distance between the NGN and KOW instead. RfaH fold-switch (Fig. 7a) requires that at least 60% of the ID contacts are broken (Fig. 7b). Following the fraction of formed native contacts for each KOW fold (Fig. 7c) reveals rugged refolding, with at least four intermediate states, I1-I4. I1 and I2 are connected to the KOW$^α$ basin, whereas I3 and I4 have a higher fraction of KOW$^β$-like native contacts (Fig. 7d). Refolding starts by the loss of native contacts at the N-terminus of α1* (residues 117-122, I1), followed by unwinding of the α2* end (148-155; I2; Fig. 7d, e). Notably, this occurs as observed in the *ops*PEC$^{Enc}$ structure (Fig. 4d and Supplementary Fig. 4e), even though the helical content of these regions was restored using homology modeling, and agrees with our hydrogen-deuterium exchange data[45]. Most ID interactions, except for those between NGN 85-100 and KOW 114-126/150-162 from the ends of α1*/α2*, persist through I2.

In I2, KOW β5* is released from NGN and interacts with β1* (average contact probability 0.9), and emergent β1*-β2* and β1*-β5*

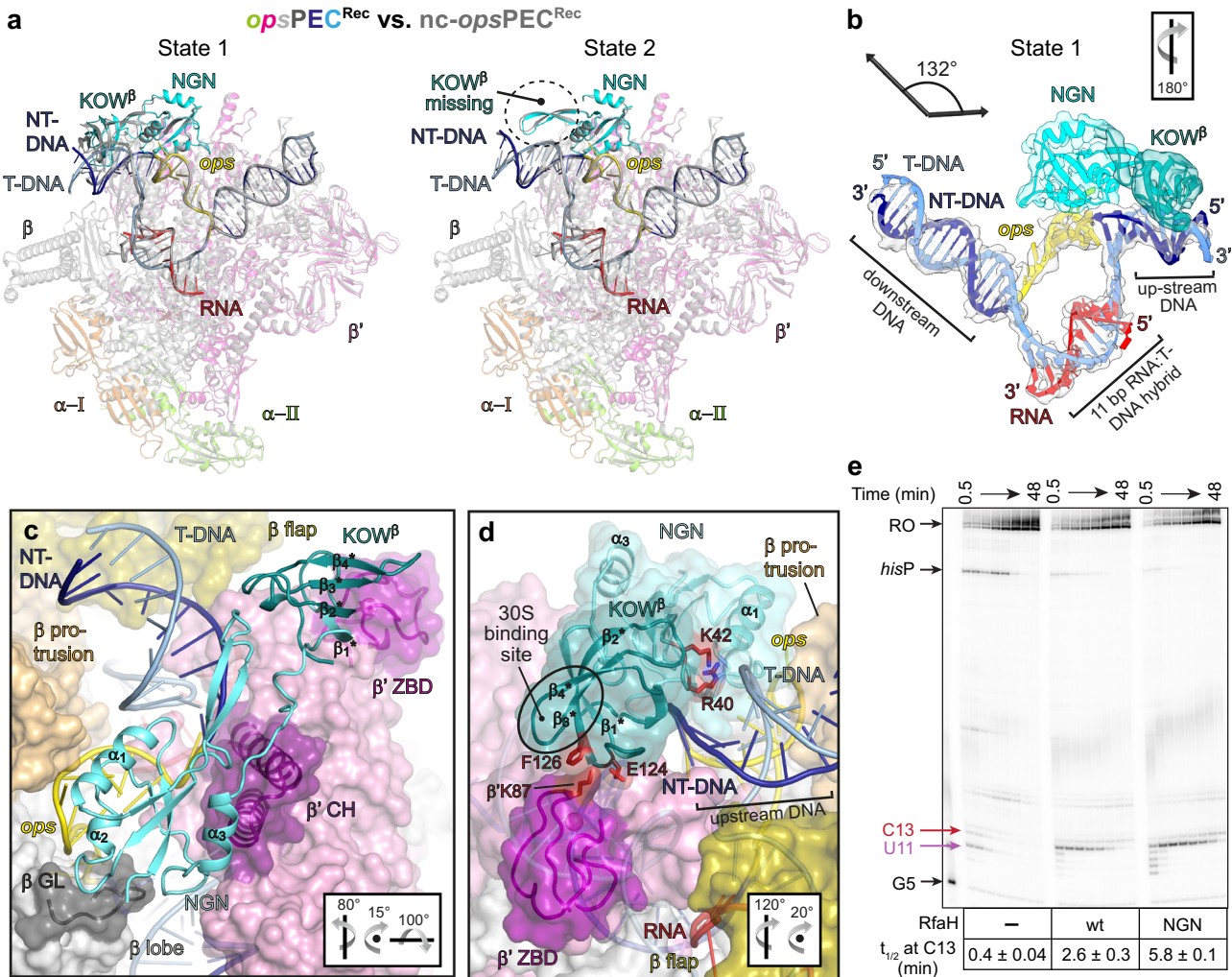

**Fig. 6 | Recruited RfaH is activated and keeps *ops*PEC in the hyper-paused state. a** Superposition of *ops*PEC^Rec (in color) and nc-*ops*PEC^Rec (gray). 3DVA of *ops*PEC^Rec cryoEM maps revealed two extrema, state 1 (left) with KOW^β bound to the β'ZBD and poorly ordered *ops*HP DNA (the latter only observed for nc-*ops*PEC^Rec) and state 2 (right) exhibiting no cryoEM density of KOW^β but well-defined *ops*HP density. **b** *ops*PEC^Rec state 1. The nucleic acids and RfaH are shown as colored cartoons along with their corresponding cryoEM density (transparent surface). The helix axes vectors of up- and downstream DNA and the angle between them are shown on top. **c, d** Positioning of RfaH within *ops*PEC^Rec. RNAP is shown as transparent surface, selected structural segments are depicted as cartoons. RfaH secondary structure elements are labeled. In (**d**), KOW^β contacts the β'ZBD and upstream DNA; side chains of interacting residues are displayed as red sticks. Orientations in (**b**–**d**) are relative to the standard view (Fig. 2a). **e** Deletion of KOW augments RfaH-induced pausing at C13. Halted radiolabelled G5 ECs were chased in the absence of RfaH or in the presence of full-length RfaH or NGN, as described in Methods. Samples withdrawn at 0.5, 1, 2, 4, 6, 12, 24, and 48 min were analyzed on a urea-acrylamide gel. The positions of *ops* G5, U11 and C13, *his*P, and run-off (RO) RNAs are indicated with arrows; for each RfaH variant (or none) the assay was repeated three times with similar results; the half-life ($t_{1/2}$) of pausing at C13 is presented as mean ± SD. Source data are provided as a Source Data file.

interactions are also observed (Fig. 7d). In I3, the probability of native β1*-β2* contacts (0.9) surpasses that of β1*-β5* (0.6) and β2*-β3* (0.4). These strands are formed just after completion of α1* unwinding and the loss of most interhelical and ID interactions, except for contacts between the KOW hairpin tip and NGN (Fig. 7d). Last, I4 is characterized by nearly complete unwinding of α2*, the loss of almost all ID interactions, and high probability native interactions between strands β1*-β5*, β2*-β3* and β3*-β4* that will later consolidate the KOW^β (Fig. 7e). The time course of KOW refolding (Fig. 7f) follows the sequential pattern, with early interactions between β1*-β5* (peak at 100 τ), followed by β1*-β2* (800 τ), and lastly β2*-β3* and β3*-β4* (1000 τ). The refolding trajectories are heterogeneous regarding the order of β1*-β5* (Supplementary Movie 1) or β1*-β2* (Supplementary Movie 2) interactions; see https://doi.org/10.5281/zenodo.10493315 for representative trajectories.

Next, we analyzed the sequence of events within RfaH and between RfaH and *ops*PEC during the transition from *ops*PEC^Enc to *ops*PEC^Rec. We examined the disruption of native ID contacts and the formation of native contacts for KOW^β, contacts between α3 (residues 90-100) and RNAP, and contacts between KOW^β and the β'ZBD as a function of time. The first event enabling RfaH refolding is the breakage of 70% of ID interactions (peak at 400 τ, Fig. 7g), in line with the evidence that the NGN-KOW contacts control RfaH metamorphosis[46,47]. Concurrently or after domain dissociation (1000 τ), α3, which harbors a highly conserved I93 residue that stabilizes the ID interface[48], binds to β'CH (Supplementary Movies 1 and 2). In these simulations, α3 gets locked in place by forming >75% of its native contacts with RNAP and enabling tighter binding of RfaH to *ops*PEC. Then, KOW refolds into the β-barrel (peak at 1100 τ). This event is not concurrent with binding to DNA, β flap or β'ZBD (Fig. 7h and Supplementary Movies 1 and 2).

These findings are in agreement with our structural (Fig. 4) and biophysical analyses[45,49] and with MD simulations performed in the absence of RNAP[46,50,51]. We conclude that the KOW transformation is independent of its interactions with the *ops*PEC.

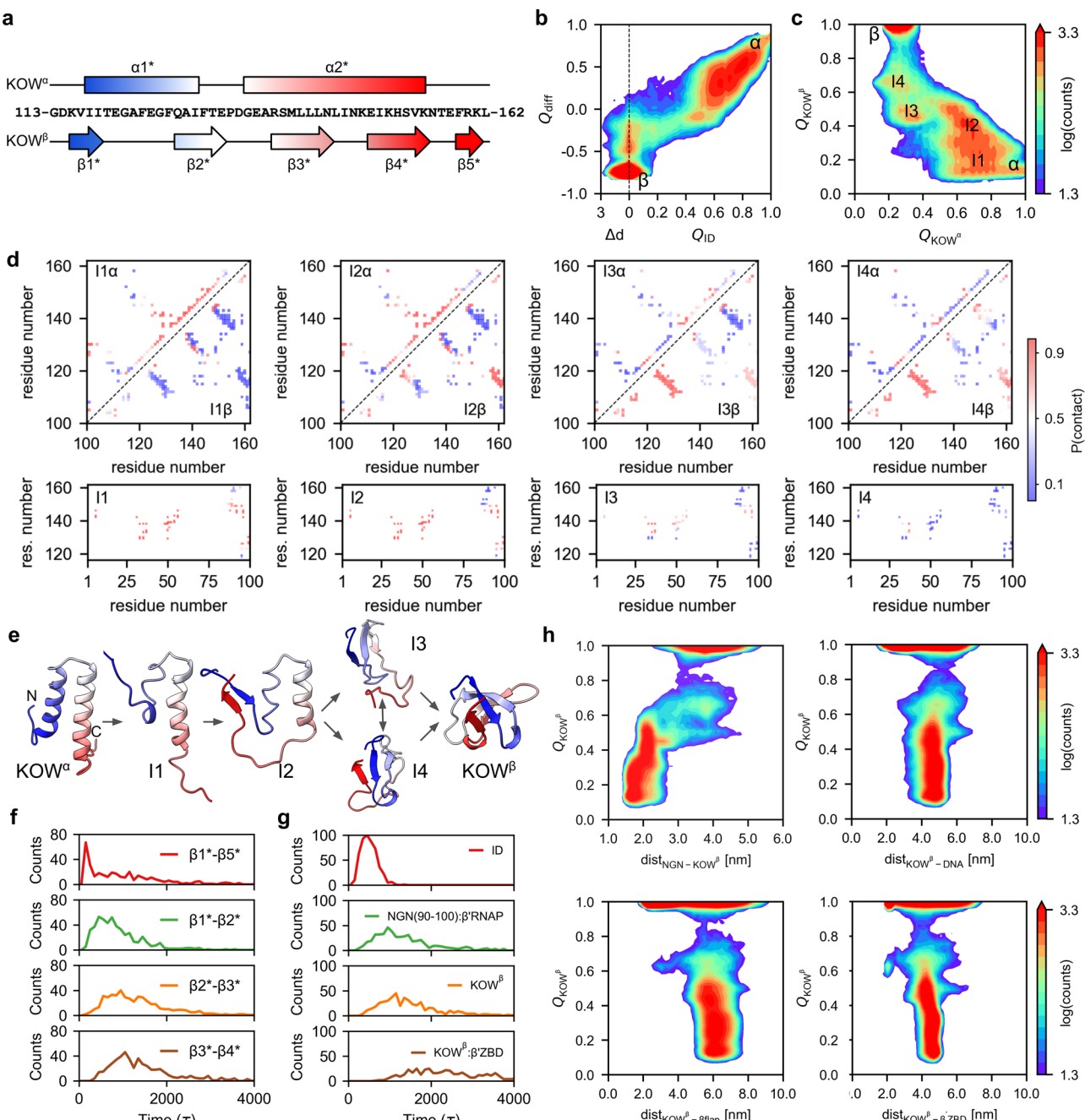

**Fig. 7 | Refolding landscape of RfaH. a** Secondary structure topology of the KOW$^\alpha$ (top) and KOW$^\beta$ (bottom). Helices are represented as rectangles and strands as arrows. **b** Refolding landscape of RfaH projected onto $Q_{ID}$ (fraction of ID contacts), $\Delta d$ (distance between domains with respect to the distance in the active state, in nm) and $Q_{diff}$ (difference in native contacts between the KOW$^\alpha$ and KOW$^\beta$). The color scheme represents the number of times each configuration is observed across all MD simulations. **c** Refolding landscape projected onto the fraction of native contacts of each KOW state ($Q_{KOW}^\alpha$ and $Q_{KOW}^\beta$). Intermediate states are labeled. **d** Probability of native contacts belonging to either KOW$^\alpha$ (upper triangle), KOW$^\beta$ (lower triangle) or ID contacts (bottom plots) present in each intermediate state. **e** Intermediates in the KOW refolding pathway. **f** Histograms of the number of β-strand formation events as a function of time. **g** Histograms of the number dissociation and association events as a function of time, with domain dissociation being the first event during refolding. **h** Landscapes of the KOW refolding as a function of the distance between KOW and NGN, DNA, β flap and β'ZBD.

## The *ops*PEC$^{Rec}$ can be arrested

Our results show that RfaH further stabilizes the already strong pause. How does *ops*PEC$^{Rec}$ resume elongation? To answer this question, we subjected *ops*PEC$^{Rec}$ after NTP addition and heating (37 °C) to cryoEM/SPA, yielding a cryoEM reconstruction at 3.0 Å resolution. The resulting complex was nearly identical to *ops*PEC$^{Rec}$ (Fig. 8a, Supplementary Fig. 6a and Supplementary Table 1) with a major exception: a clear additional density in the secondary channel showed that the transcript

had been elongated by at least two nts followed by RNAP backtracking (Fig. 8b, c). These findings establish that *ops*PEC$^{Rec}$ is elongation-competent (i.e., can transiently adopt a post-translocated state) and that, following nucleotide addition, RNAP slides back, generating *ops*PEC$^{Back}$.

In the cell, backtracked RNAP can be rescued by Gre factors[41], Mfd[52], or the coupled ribosome[53]. We next tested if RNAP escape from *ops* can be promoted by anti-backtracking factors in vitro. In

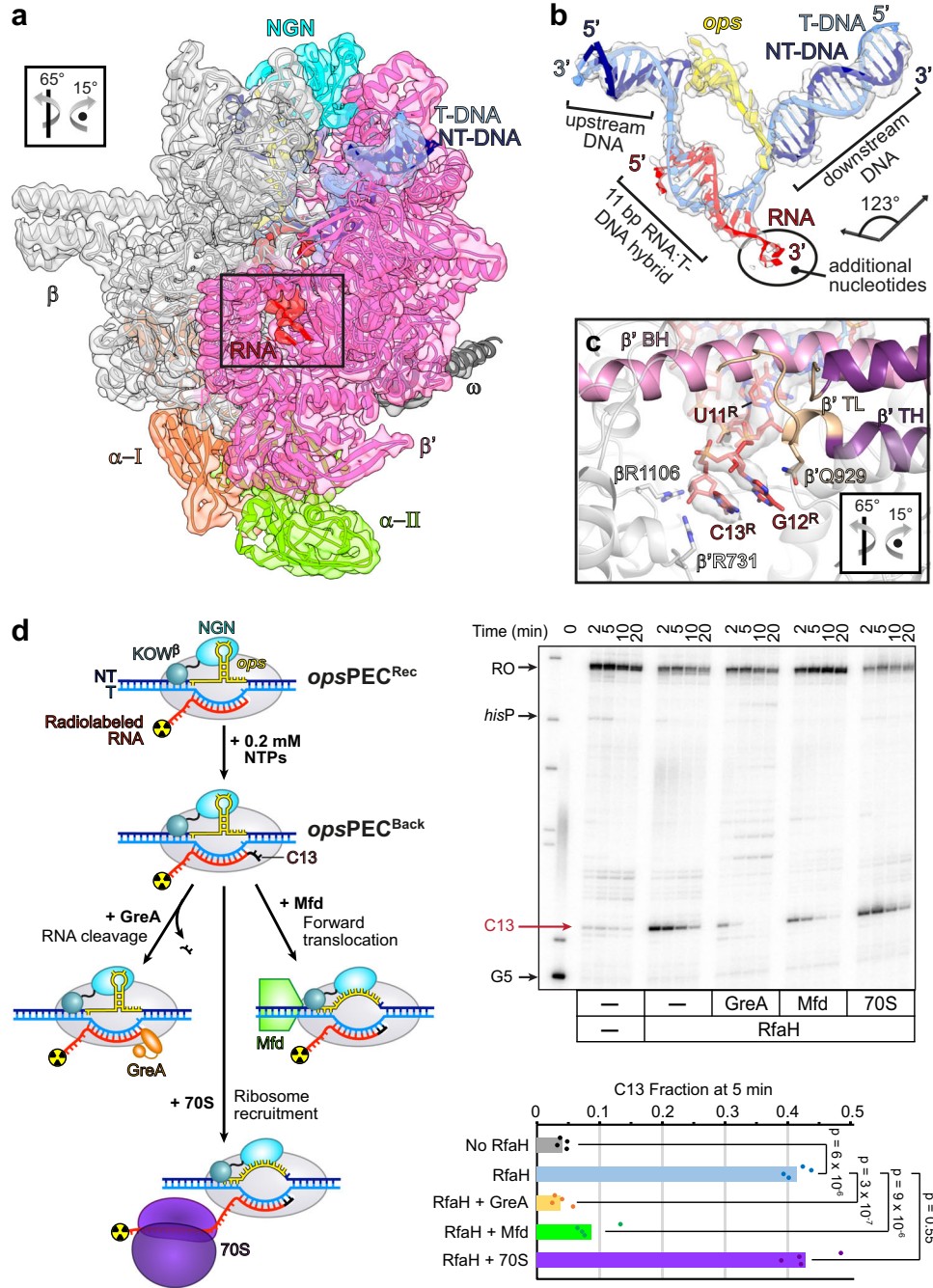

**Fig. 8 | RNAP escape from *ops* is hindered by backtracking and requires auxiliary factors. a** Upon nucleotide addition, *ops*PEC^Rec forms *ops*PEC^Back (map as transparent surface, model as cartoon) with two additional RNA nucleotides in the secondary channel (boxed). **b** CryoEM map and model of the *ops*PEC^Back nucleic acid scaffold. Additional 3′ RNA nts resulting from RNA extension and subsequent backtracking are indicated. **c** Close-up view of the region boxed in (**a**). RNAP, β′BH and β′TL are shown as cartoons, RNA as sticks along with the corresponding cryoEM density. RNAP side chains contacting the protruding RNA are shown as sticks. Orientations in (**a**) and (**c**) are relative to the standard view (Fig. 2a). **d** The

arrested *ops*PEC^Back can be rescued by either GreA-assisted RNA cleavage or Mfd-medicated forward translocation. Halted radiolabelled G5 ECs were chased with 0.2 mM NTPs in the absence or in the presence of indicated proteins. Samples withdrawn at 2, 5, 10, and 20 min were analyzed on a urea-acrylamide gel. The positions of *ops* G5 and C13, *his*P, and run-off (RO) RNAs are indicated. RNA fractions at C13 after 5 min incubation with NTPs were calculated from four independent datasets. Raw data points (scattered dots) and the mean values are shown. Source data are provided as a Source Data file.

agreement with our structural data, RfaH strongly delays elongation two nts downstream from the *ops* pause site: the C13 pause persists for minutes even at 0.2 mM NTPs (Fig. 8d). The addition of GreA, which induces RNA cleavage in backtracked ECs[41], dramatically shortened the pause. A similar but less dramatic effect was observed with Mfd, a DNA translocase that pushes RNAP forward[52]. By contrast, the 70 S ribosome did not promote escape, an expected result given that the

ribosome must exert force on RNAP to assist forward translocation, not just sterically block reverse translocation[53], and translation initiates only 50+ nts downstream from the *ops* site.

## The KOW domain contributes to pause escape
3DVA of *ops*PEC^Rec revealed that KOW^β binding at β′ZBD was correlated with a movement of the upstream DNA towards the β protrusion and

flap (Supplementary Fig. 6 and Supplementary Movies 3 and 4). This suggests that KOW$^\beta$ reinforces HL-DNA contacts, which are already established in opsPEC$^{Enc}$, and may either support hyper-pausing by helping drive the proximal end of the upstream duplex into the β′ rudder or, alternatively, counteract backtracking of opsPEC$^{Rec}$. The removal of KOW potentiates the RfaH-induced delay at C13 (Fig. 6e). While a fraction of arrested complexes may escape by releasing RfaH and reformation of the autoinhibited state in full-length RfaH, but not in the isolated NGN, this observation is consistent with the idea that, after RfaH accommodation and refolding, KOW$^\beta$/HL act as anti-backtracking devices. A similar effect has been observed for SuhB-reinforced NusG-upstream DNA contacts in an rRNA antitermination complex[6].

3DVA also showed that in some complexes, density for KOW$^\beta$ was anti-correlated with density for the opsHP; i.e., one boundary state exhibited clear density for KOW$^\beta$ bound at β′ZBD but weak density for the opsHP and downstream ops nts, whereas the other lacked density for KOW$^\beta$ but exhibited very well-defined density for the opsHP and downstream nts (Supplementary Movies 3 and 4). Thus, in addition to dampening backtracking, KOW$^\beta$ binding at β′ZBD seems to facilitate pause escape by destabilizing opsHP-NGN interactions.

### NusA and KOW may cooperate during ribosome loading

NusA is a general elongation factor that associates with most ECs[3] to modulate RNAP pausing, termination, and antitermination through contacts to the RNA or accessory factors[6,37,54–56]. NusA binds to the βFTH[37] and is expected to associate with, and possibly trigger conformational changes in, RfaH-bound ECs. To ascertain that our conclusions would hold in the presence of NusA, we assembled a NusA-modified opsPEC$^{Rec}$ and determined its structure at 3.2 Å resolution (Supplementary Fig. 7b). The cryoEM reconstruction revealed that, except for added NusA, this complex is essentially identical to opsPEC$^{Rec}$, suggesting that NusA binding does not induce changes in opsPEC$^{Rec}$ (Supplementary Fig. 7c–f) or alter the KOW$^\beta$ presentation. NusA can aid in coupling transcription to translation[21] and our data show that NusA can interact with RNAP and the ribosome in the RfaH-modified EC. Thus, NusA could cooperate with the RfaH KOW domain to provide additional docking sites for the pioneering ribosome.

## Discussion

Early findings that RfaH binding to the transcribing RNAP requires sequence-specific contacts to the NT DNA[32] and dramatic structural rearrangements[12,34] prompted us to propose a model for co-transcriptional recruitment and activation of RfaH. Here, we report structural and in silico data that support and extend this model (Fig. 9). First, we present a structure of E. coli RNAP paused at ops, an archetypal regulatory site that recruits RfaH. Most notably, the structure reveals an 8-nt-long NT DNA hairpin (opsHP) that triggers RNAP swiveling to stabilize a catalytically inactive pre-translocated state while displaying multiple recognition motifs for RfaH. Second, we present a structure of a highly transient, yet functionally crucial, encounter complex that captures autoinhibited RfaH bound to opsPEC. In this structure, still autoinhibited RfaH binds to, and repositions, the opsHP to initiate expansion of the transcription bubble and the RNA:DNA hybrid, stabilizing the paused state. Third, we present a structure of RfaH fully engaged with RNAP in an even deeper paused state, which is likely necessary to recruit the pioneering ribosome. Fourth, we present a structure that captures an unsuccessful attempt of RfaH-bound RNAP to escape the recruitment site, leading to a backtracked state which is rescued by accessory factors.

Pausing at ops is a prerequisite for RfaH recruitment[12] and ops is one of the strongest pauses in E. coli[39], as could be expected because a failure of RfaH engagement compromises the cell wall integrity[57]. The opsPEC structure reveals a unique geometry, a canonical 10-bp hybrid but 11-nt long ss NT DNA that forms the hairpin stabilized by a

multitude of positively-charged RNAP residues (Fig. 3a). The two downstream nts, G10 and T11, stack on each other and are further stabilized by βW183, whereas G12 is flipped into the CRE pocket, leaving C12$^T$ free to contact βR542 (Fig. 3d). The unfolded TL is trapped by salt bridges between the TL β′R933 and the fork loop βE546/D549 (Fig. 2c) and the swivel module rotates by 5.8 ° (Fig. 2b), stabilized by opsHP-β lobe interactions. Recently, NusG-dependent pausing in Gram-positive bacteria was suggested to rely on a similar mechanism wherein the NusG-bound NT DNA is placed in a cleft between NusG and β lobe, blocking RNAP return to the non-swiveled state[58]. Strikingly, through contacts with RNAP, the opsHP is able to lock the swivel module in the absence of additional inputs. Comparison of opsPEC, an exemplar of a consensus pause, to other PECs offers insights into the sequence-structure relationships in pausing (Supplementary Discussion).

We show that the preformed opsHP recruits the autoinhibited RfaH$^{CC}$ through contacts to G5 and T6 (Fig. 4f). Many ops-like hairpins, which differ only in these loop residues and are thus expected to induce pausing, are encoded in the NT strand of MG1655 operons; RfaH is recruited only to sites that have a T at position 6[29], but these comprise fewer than half of ops-like sequences. Do NT-strand hairpins that have A, G or C in the loop recruit, or perhaps exclude, other modulators of elongation? Housekeeping NusGs from Bacillus subtilis[58] and Mycobacterium tuberculosis[59] interact with the NT DNA and B. subtilis NusG displays a preference for T-tracks[60]. E. coli NusG makes no contacts to DNA[13] and readily dissociates from the EC[61]. NusG is excluded from the EC by RfaH[13,29] and, conceivably, by NT-DNA structures. By contrast, RfaH orthologs must avoid recruitment to "wrong" sites, and the NT DNA readout provides means for selectivity.

We show how autoinhibited RfaH sets the stage for the final NGN placement, with three elements making contacts to the opsHP, β′CH and β gate loop. In the encounter complex (Fig. 9c), NGN cannot be fully accommodated between β lobe, β protrusion, and β′CH, and swiveling is required to fit the additional bulk of KOW$^\alpha$ between these RNAP elements. While anchored at the upstream DNA, NGN twists and repositions the opsHP, leading to melting of one bp in the upstream DNA and the formation of an overextended 11-bp hybrid, which counteracts translocation and stabilizes the pause. RfaH$^{CC}$ clamps the NT strand, hindering the return to the non-swiveled state and strengthening the pause. Contacts with opsHP position NGN near the β′CH tip, which wedges between the RfaH domains, initiating displacement of KOW$^\alpha$ (Fig. 4d). Consequently, RfaH drives the wedge further and further between KOW and NGN until they dissociate completely. Thus, the opsHP loop has two functions: (i) sequence-specific recognition by RfaH and (ii) anchoring RfaH to initiate its activation.

The opsPEC$^{Rec}$ structure shows that, once unmasked, α3 packs against the β′CH, the principal RfaH-binding site on RNAP. The opsHP:NGN interactions are identical to those in opsPEC$^{Enc}$, while the region around T73 contacts the βGL, the second RNAP-binding site (Figs. 6c and 9d). The NGN-RNAP interactions account for the remarkable stability of RfaH-EC contacts throughout transcription and for the anti-pausing activity of RfaH in vitro[12,13,28] but make only a small contribution to its overall effect on gene expression[15]. The KOW binding to ribosome is critical for RfaH activity and is, in turn, dependent on refolding of the liberated KOW$^\alpha$ into a NusG-like β-barrel that creates a contact surface for S10[34].

KOW refolds spontaneously when freed from NGN[34,35], but the fold-switch could be altered in the context of the opsPEC. Our MD simulations reveal that refolding starts from the ends of the α-helical KOW hairpin (Fig. 7) consistent with its partial unfolding in the opsPEC$^{Enc}$. Once α3 is released and gets locked in place, KOW refolds into the five-stranded β-barrel (Supplementary Movie 2), largely as observed with RfaH in isolation[43]. We conclude that all the information required for KOW transformation is encoded in its primary sequence,

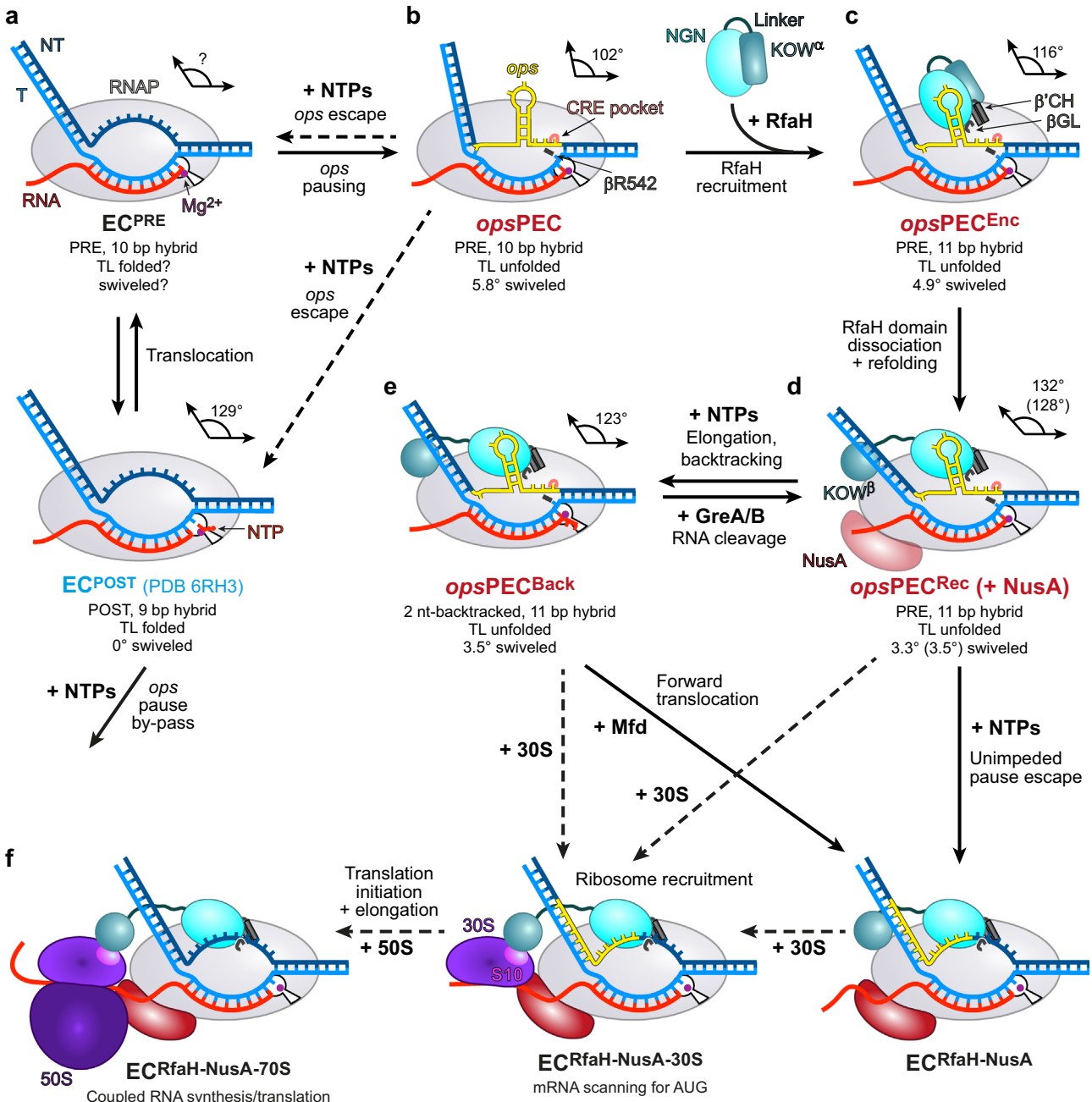

**Fig. 9 | Model of co-transcriptional recruitment and activation of RfaH. a** During processive RNA synthesis, RNAP adds nucleotides one by one to the 3′-end of the growing RNA. Following nucleotide addition, the NTP in the active site turns into the RNA 3′-end and the 9-bp RNA:DNA hybrid converts into the 10-bp hybrid. RNAP must then translocate by 1 nt, melting 1 bp of the downstream DNA to place the next T-DNA base in the active site, ready to accept the incoming NTP; a failure to translocate leads to pausing. **b** The *ops* element forms a short NT-DNA hairpin exposed on the RNAP surface. The *ops*HP interactions with RNAP push upstream DNA into the β′ zipper and β′CH, thereby supporting RNAP swiveling, and stabilize a pre-translocated state with a 10-bp hybrid and extended bubble. **c** The auto-inhibited RfaH, in which the RNAP-binding site on NGN is masked by KOWα, docks near its final binding site using the *ops*HP as an anchor to form an encounter complex, in which NGN grasps and twists the *ops*HP to hyper-stabilize the PEC with an 11-bp hybrid. **d** Upon domain dissociation, NGN takes its final position whereas KOW refolds into the β-barrel. **e** The *ops*PEC^Rec can extend the RNA but, unable to break NGN-*ops*HP interactions, backtracks to the original position. Escape is facilitated by anti-backtracking factors GreA and Mfd. **f** By extending the lifetime of the *ops* pause, RfaH may favor recruitment of the ribosome. After escape, RfaH will promote ribosome scanning the mRNA for a start codon, where the translation initiation complex assembles, and coupling thereafter. Complexes with previously known structures are labeled in blue, complexes with structures determined in this study in red, and hypothetical complexes in black.

and that its contacts to DNA and RNAP are established following the fold-switch. Further work is required to fully explore the steps of NGN accommodation into its binding site and subsequent fold-switching.

RfaH hyper-stabilizes the *ops* pause, an effect opposite to its pause-suppressing activity at any other site[32]. We show that *ops*PEC^Rec backtracks to the original position upon nt addition (Fig. 9e),

reminiscent of PECs stabilized by the initiation σ-factor, which can extend RNA and the bubble, scrunching both DNA strands to maintain contacts to the −10-like NT DNA[62]. It is hypothesized that the energy stored in the scrunched strands is used to break the σ-DNA contacts to overcome the pause[62]. We do not observe scrunching, but the transition to *ops*PEC^Back must be accompanied by at least two translocation

steps and likely requires scrunching, as proposed earlier[63]. The energy stored in the compressed bubble/hybrid and hypothetical scrunched states could be used to break the NGN/NT contacts and melt the *ops*HP to overcome the pause.

Failure to load the ribosome abolishes the expression of RfaH-controlled genes[23]. We hypothesize that a ribosome is recruited at the *ops* site through direct contacts to KOW$^{\beta}$, and that backtracking extends the time window for this recruitment. By bridging RNAP and ribosome, RfaH will promote ribosome scanning toward a start codon and coupling thereafter. It remains to be determined whether the 30 S or 70 S ribosome is recruited by RfaH and when and whether RNA is required during the initial loading.

Collectively, our results show that the information encoded in the *ops* and RfaH sequences fully controls every step in the RfaH cycle—from RNAP pausing and RfaH loading (*ops*) to RfaH activation and finally ribosome recruitment (RfaH). The KOW transformation that activates RfaH, initially thought to be unique to that protein, has recently been discovered to be ancient and ubiquitous[64], arguing that our insights into the mechanisms of recruitment and metamorphosis of RfaH would be applicable to NusG homologs across all life.

## Methods

### Molecular cloning

Plasmid pET19mod_*rfaH-F51C-S139C* encoding RfaH$^{CC}$ was generated by successive site-directed mutagenesis according to the QuickChange Site-Directed Mutagenesis Kit protocol, using pET19mod_*rfaH* as a template[33]. All other plasmids were constructed by standard molecular biology approaches with restriction and modification enzymes from New England Biolabs. Sequences of all plasmids were confirmed by Sanger sequencing either at the Genomics Shared Resource Facility, Ohio State University, USA or at Eurofins Genomics, Germany.

### Protein production and isotopic labeling

*E. coli* BL21 (DE3) strain was used for overexpression of all target genes. Antibiotics were added into LB medium when needed as follows: carbenicillin 100 μg/mL, kanamycin 50 μg/mL. If not stated otherwise protein concentrations were determined by measuring the absorbance at 280 nm ($A_{280}$) on a Nanodrop ND-1000 spectrometer or a BioSpectrometer Basic (Eppendorf). The quality of all proteins used for structural studies was checked according to the guidelines established by ARBRE-MOBIEU and P4EU (https://arbre-mobieu.eu/guidelines-on-protein-quality-control)[65]. In brief, purity was checked by sodium dodecyl sulfate gel electrophoresis (SDS-PAGE), the absence of nucleic acids by UV spectroscopy, the identity by mass spectroscopy and/or NMR spectroscopy, the folding state by CD and/or NMR spectroscopy, and the homogeneity as well as the absence of aggregates by analytical gel filtration. The purity of proteins used in pause assays was ensured by SDS-PAGE.

### RNAP

Production of RNAP for cryoEM experiments was based on ref. 33. Briefly, *E. coli* BL21 (DE 3) cells harboring pVS10 (encodes *E. coli* RNAP subunits α, β, β' with C-terminal His$_6$-tag, and ω[66] were grown at 37 °C in LB medium supplemented with 100 μg/ml ampicillin to an optical density at 600 nm ($OD_{600}$) of ~0.6. The temperature was then lowered to 16 °C and gene expression was induced 30 min later by addition of 0.5 mM isopropyl-1-thio-β-D-galactopyranoside (IPTG). Overexpression was performed overnight at these conditions and the cells were collected the next morning by centrifugation. The pellets were then resuspended in buffer A$^{RNAP}$ (50 mM Tris/HCl, 500 mM NaCl, 5 % (v/v) glycerol, 1 mM β-mercaptoethanol (β-ME), pH 6.9) supplemented with 10 mM imidazole, DNase I, and ½ tablet of protease inhibitor (cOmplete, EDTA-free) and subsequently lysed using a microfluidizer. The soluble fraction was then loaded onto a Ni$^{2+}$-Chelating Sepharose column (40 ml column volume (CV)), the column was washed with 6 CV of buffer A$^{RNAP}$ containing 10 mM imidazole, and RNAP was subsequently eluted using the following gradient: 10 mM to ~110 mM imidazole over 4 CV, 4 CV of ~110 mM imidazole, 3 CV of ~260 mM imidazole (all in buffer A$^{RNAP}$). RNAP-containing fractions were pooled and dialyzed against buffer B$^{RNAP}$ (50 mM Tris/HCl, 0.5 mM EDTA, 5 % (v/v) glycerol, 1 mM β-ME, pH 6.9) containing 50 mM NaCl and the solution was then applied to two coupled 5 ml HiTrap Heparin HP columns (CV = 10 ml). The columns were washed with 15 CV of buffer B$^{RNAP}$ supplemented with 280 mM NaCl and the protein was subsequently eluted using a constant gradient from 280 mM to 550 mM NaCl in buffer B$^{RNAP}$. The eluate, containing a mixture of mainly core-RNAP and traces of holo-RNAP, was dialyzed against buffer C$^{RNAP}$ (20 mM Tris/HCl, 0.5 mM EDTA, 5% (v/v) glycerol, 1 mM β-ME, pH 7.9) supplemented with 100 mM NaCl, then concentrated by ultrafiltration and finally applied to a 1 ml MonoQ 5/50 FPLC column. The column was then first washed with 10 CV of 100 mM NaCl, followed by 10 CV of 280 mM NaCl in buffer C$^{RNAP}$ and then subjected to a constant NaCl gradient from 280 mM to ~440 mM in buffer C$^{RNAP}$ over 20 CV to separate core- and holo-enzymes, respectively. Fractions containing core-RNAP were then dialyzed against buffer D$^{RNAP}$ (50 mM Tris/HCl, 200 mM NaCl, 0.5 mM EDTA, 5 % (v/v) glycerol, 1 mM β-ME, pH 6.9), subsequently concentrated by ultrafiltration, and then subjected to size exclusion chromatography (SEC) using a HiLoad Superdex 200 16/600 column to remove aggregates. Fractions containing pure, homogeneous enzyme were concentrated, supplemented with glycerol to 20% (v/v), aliquoted and then flash frozen via liquid nitrogen and stored at −80 °C until further usage. The final concentration of RNAP was ~20 μM. A simple in vitro transcription assay based on extension of a fluorescent RNA primer[67] was used to confirm the enzyme activity. RNAPs (wt or mutant) for pause assays were purified using established protocols[66]; for RNAP mutants, protein expression was induced overnight with 0.2 mM IPTG at 16 °C.

### Wild-type RfaH

Production of wt RfaH for NMR-spectroscopy and cryoEM experiments was done as in[33]. *E. coli* BL21 (DE3) cells harboring plasmid pET19bmod_*rfaH* (encoding wt RfaH with N-terminal His$_6$-tag followed by a Tobacco Etch Virus [TEV] cleavage site) were grown at 37 °C in M9 medium supplemented with 100 μg/ml ampicillin to an $OD_{600}$ of 0.5–0.6. The temperature was then reduced to 20 °C, gene expression was induced 30 min later by the addition of 0.2 mM IPTG and carried out over-night under these conditions. Cells were harvested by centrifugation the next day. For purification, the cells were resuspended in buffer A$^{RfaH}$ (50 mM Tris/HCl, 300 mM NaCl, 5 % (v/v) glycerol, 1 mM dithiothreitol (DTT), pH 7.5), supplemented with 10 mM imidazole, DNase I and ½ tablet of protein inhibitor cocktail (cOmplete, EDTA-free) and lysed using a microfluidizer. The supernatant was then loaded onto a 5 ml HisTrap HP column, the column was washed with 10 CV buffer A$^{RfaH}$ containing 10 mM imidazole, and RfaH was eluted using an imidazole step gradient ( ~ 60 mM, ~110 mM, ~160 mM, ~210 mM, ~310 mM; all in buffer A$^{RfaH}$, 5 CV per step). RfaH-containing fractions were pooled and dialyzed for 1 h against buffer A$^{RfaH}$ before the addition of the TEV-protease to the dialysis membrane. Cleavage was carried out over-night at 4 °C. The dialysate was then applied to a 5 ml HisTrap HP column again, and the column was washed with buffer A$^{RfaH}$ containing 10 mM imidazole. The cleaved RfaH was largely present in the wash fraction rather than in the flow-through, presumably due to unspecific binding to the column material. The corresponding fractions were then concentrated to 250 μM by ultrafiltration, aliquoted and flash-frozen in liquid nitrogen. The samples were stored at −80 °C until further usage. The absence of aggregates from the preparation was confirmed by analytical SEC using a Superdex 75 Increase 10/300 GL column (Cytiva, CV = 24 ml; Supplementary Fig. 3b). Full-length wt RfaH and the isolated NGN for pause assays were prepared as in[12].

## RfaH$^{CC}$

When expressed in *E. coli*, we found that RfaH$^{CC}$ was either largely insoluble or incorrectly folded under all expression conditions and for all strains tested, thus hampering structural studies. We therefore chose the strategy of refolding the protein isolated from inclusion bodies. Expression was carried out as for wt RfaH but using plasmid pET19bmod_*rfaH-F51C-S139C* instead. For purification, the resulting cell pellets were resuspended in buffer A$^{RfaH}$ containing 5 mM DTT and 10 mM imidazole, supplemented with DNase I and ½ tablet protease inhibitor cocktail (cOmplete, EDTA-free) and then lysed using a microfluidizer. The insoluble fraction was collected by centrifugation and washed by extensive stirring with each 40 ml of (i) buffer B$^{RfaH}$ (10 mM EDTA, 1 mg/ml deoxycholate, 10 mM β-ME, pH 8.0) containing 8 mg of lysozyme, (ii) Buffer B$^{RfaH}$, (iii) 2x buffer C$^{RfaH}$ (50 mM Tris/HCl, 1 M NaCl, 10 mM EDTA, 10 mM DTT, pH 7.5) and (iv) H$_2$O to isolate the inclusion bodies. The protein was then solubilized and unfolded by stirring the pellet at room temperature in buffer D$^{RfaH}$ (50 mM Tris/HCl, 500 mM NaCl, 8 M urea, 5 mM β-ME, pH 7.5) for at least 2 h. The solution was subsequently dialyzed against buffer E$^{RfaH}$ (50 mM Tris/HCl, 1 M NaCl, L-glutathione (1 mM reduced, 0.1 mM oxidized), pH 7.5) over-night at 4 °C to allow refolding of RfaH$^{CC}$. The protein remaining soluble was then purified by Ni$^{2+}$ affinity chromatography similarly to wt RfaH, except that reducing agents were omitted from the buffers. TEV-cleavage was also performed as for the wt protein, but the buffer A$^{RfaH}$ used for dialysis and proteolysis contained 0.05 mM DTT to avoid potential destruction of the disulfide bridge but still support sufficient TEV activity. The cleaved protein was then applied to a 5 ml HisTrap HP column, the column was washed with 5 CV buffer A$^{RfaH}$ containing 10 mM imidazole but no DTT, and the flow-through/washing fractions were concentrated by ultrafiltration. The samples containing 50 – 100 µM of RfaH$^{CC}$ were subsequently aliquoted, flash-frozen in liquid nitrogen and then stored at −80 °C until further usage. The absence of aggregates from the final samples was confirmed by analytical SEC using a Superdex 75 Increase 10/300 GL column (Cytiva, CV = 24 ml; Supplementary Fig. 3b).

For most RfaH$^{CC}$ preparations obtained by the above procedure, the C51-C139 disulfide bridge was at least in part reduced (i.e., open), as indicated by NMR spectroscopy or by the shift between the bands of reduced and oxidized RfaH$^{CC}$ samples on an SDS gel (Fig. 4a)[12], respectively. To obtain a closed disulfide bridge, 100 µM Cu$^{II}$ phenanthroline was added, the sample was incubated for 1 min at room temperature and the reaction was stopped by the addition of 1 mM EDTA. The buffer was then immediately exchanged by either SEC or via a PD Minitrap G25 desalting column.

## NusA

Production of NusA was based on ref. 68. *E. coli* B21(DE3) cells harboring pTKK19 nusA(1-495) were grown at 37 °C. Overexpression was induced at $OD_{600}$ ˜ 0.7 by the addition of IPTG (final concentration: 1 mM). Cells were harvested after 4 hours by centrifugation (9000 × *g*, 15 min, 4 °C). The cell pellet was resuspended in buffer A$^{NusA}$ (20 mM Tris/HCl, pH 7.9, 500 mM NaCl, 5 mM imidazole, 1 mM β-ME) and lysed using a microfluidizer (Microfluidics, Newton, MA, USA). Following centrifugation (12,000 × *g*, 30 min, 4 °C) the crude extract was applied to a 5 ml HisTrap HP column (Cytiva, Munich, Germany). The column was washed with buffer A$^{NusA}$ and elution was performed using a step gradient from 5 mM to 1 M imidazole in buffer A$^{NusA}$. Fractions containing the His$_{10}$-NusA fusion protein were combined and the target protein was cleaved by PreScission protease during overnight dialysis (4 °C) against buffer B$^{NusA}$ (20 mM Tris/HCl, pH 8, 1 mM β-ME). The protein solution was applied to a 5 ml GSTrap FF column (Cytiva, Munich, Germany) and the flow-through was loaded onto a 5 ml QXL column (Cytiva, Munich, Germany) subsequently. Elution of NusA was carried out using a step gradient from 0 to 1 M NaCl in buffer B$^{NusA}$. Fractions containing pure NusA were combined, dialyzed against 5 l

20 mM Tris/HCl, pH 7.5, 50 mM NaCl, 1 mM dithiothreitol (DTT) and concentrated using ultrafiltration units (Viva Science, molecular weight cut-off (MWCO): 10 kDa). Finally, aliquots were shock frozen in liquid nitrogen and stored at −80 °C.

**σ70**. A published protocol was used for the production of *E. coli* σ$^{70}$[66].

**GreA**. A published protocol was used for the production of *E. coli* GreA[69].

**Mfd**. A published protocol was used for the production of *E. coli* Mfd[70].

**70S ribosome**, prepared from *E. coli* MRE-600, is a gift from Kurt Fredrick, The Ohio State University.

**Isotopic labeling**. For $^{15}$N-labeling, cells were grown in M9 medium[71] containing ($^{15}$NH$_4$)$_2$SO$_4$ (Sigma/Merck KGaA, Germany) as sole nitrogen source. Expression and purification strategies were identical to the production protocols of unlabeled proteins.

## CD spectroscopy

CD spectra of RfaH variants were acquired at a Jasco J-1100 spectro-polarimeter (Jasco Deutschland GmbH, Pfungstadt, Germany) at 25 °C in continuous scan mode (scan speed: 50 nm/min) with a step width of 0.1 nm, using a 1 mm quartz cuvette (Hellma GmbH & Co. KG, Müll-heim, Germany). All samples contained ˜10-12 µM (˜0,18-0,2 mg/ml) of protein in CD buffer (10 mM K phosphate, pH 7.0). RfaH$^{CC}$ was measured in CD-buffer (i.e. in the oxidized state) and in CD-buffer containing 0.5 mM of the reducing agent tris (2-carboxyethyl) phosphine (TCEP). The resulting spectra were smoothed mathematically using a Savitzky-Golay filter and then normalized to protein concentration (*c*, in mM), number of residues (*N*) and length of the light path (*d*, in cm) to obtain the mean residue weight ellipticity ($Q_{MRW}$) by Eq. (1):

$$\Theta_{MRW} = \frac{100 \cdot \theta}{N \cdot c \cdot d} \qquad (1)$$

The concentration of each sample was determined via Beer-Lambert's law from the absorption at 205 nm as measured by the CD spectrometer and using the theoretical molar extinction coefficient at 205 nm obtained from the Protein A205 Calculator (https://www.gmclore.org/clore/Software/A205.html)[72].

## NMR spectroscopy

NMR data were collected on Bruker Ascend Aeon 900 and Ascend Aeon 1000 spectrometers ($B_0$ = 21.1 T and 23.5 T, respectively) equipped with TCI CryoProbes (inverse $^1$H, $^{13}$C, $^{15}$N triple resonance probes). $^{15}$N-wt RfaH (100 µM) was in 10 mM K phosphate, 10 % (v/v) D$_2$O, pH 7.0, while $^{15}$N-RfaH$^{CC}$ samples (50 µM) were buffered by 50 mM Na phosphate, 50 mM KCl, 0.3 mM EDTA, 10 % (v/v) D$_2$O, pH 7.5. All measurements were carried out at 288 K using 5 mm NMR-tubes. Standard two-dimensional (2D) [$^1$H, $^{15}$N]-heteronuclear single quantum coherence (HSQC) spectra[73] were acquired using TopSpin (version 3.5; Bruker) at a spectral window of 119 ± 12 ppm for $^{15}$N and 4.7 ± 6 ppm for $^1$H. The spectra were referenced externally using an Na trimethylsilyl-propanesulfonate (DSS) containing standard sample (2 mM Sucrose, 0.5 mM DSS, 2 mM NaN$_3$ in 90 % H$_2$O, 10 % D$_2$O; Bruker). Data were processed using in-house written programs that yield the same information as publicly available programs such as NMRPipe[74], and visualized using NMRViewJ (version 9.2.b20, One Moon Scientific, Inc., Westfield, NJ, USA). The corresponding peak assignments for wt *E. coli* RfaH were taken from a previous study[34] (BMRB accession number 52345).

## Pause assays

Linear templates for in vitro transcription were generated by PCR amplification (Supplementary Table 3) and purified using QIAquick PCR purification kit. To form RNAP holoenzyme, the RNAP core was

mixed with σ[70] at 1:3 molar ratio and incubated for 15 min at 30 °C. For all pause assays, halted complexes were formed by incubating linear DNA template (30 nM), RNAP holoenzyme (40 nM; wt or a mutant variant), ApU (100 μM), and a starting NTPs in TGA2 (20 mM Tris-acetate, 20 mM Na-acetate, 2 mM Mg-acetate, 5% glycerol, 1 mM DTT, 0.1 mM EDTA, pH 7.9) for 15 min at 37 °C. For pause assay with RNAP and RfaH variants, halted ECs were formed on pIA349-derived DNA template with 1 μM GTP, 5 μM ATP, 5 μM CTP, and 0.1 μCi/μL [α-32P]-GTP. Transcription was restarted by the addition of ChaseG (final concentrations: 10 μM GTP, 200 μM each ATP, CTP, and UTP, and 25 μg/mL rifapentine).

For assays with RfaH, full-length RfaH or the isolated NGN were added into the halted complex to 50 nM (or an equal volume of storage buffer), followed by a 3 min incubation at 37 °C. Transcription was then restarted by the addition of ChaseG.

For assays of RNAP escape from the *ops* site, halted ECs were formed on pIA1633-derived template with ApU and starting NTPs (1 μM ATP, 5 μM GTP, 5 μM CTP, and 0.1 μCi/μL [α-32P]-ATP). Then 50 nM of wt RfaH was added and, after 3 min 37 °C, 1 μM GreA/Mfd/70S was introduced into the mixture. The reaction was allowed to sit for 1 min at 37 °C, followed by the addition of Chase200 (200 μM each ATP, CTP, GTP, and UTP, 1 mM dATP, and 25 μg/mL rifapentine).

Samples were removed at time points indicated in the figures and quenched by the addition of an equal volume of STOP buffer (10 M urea, 60 mM EDTA, 45 mM Tris-borate, pH 8.3, 0.1% bromophenol blue, and 0.1% xylene cyanol). Samples were heated for 2.5 min at 95 °C and separated by electrophoresis in denaturing 9% acrylamide (19:1) gels (7 M urea, 0.5× TBE). The gels were dried and RNA products were visualized using the FLA9000 Phosphorimaging System and quantified using ImageQuant software (version 5.2; Cytiva). The *ops* pause half-life was calculated as described in ref. 75.

## CryoEM: assembly of *ops*PECs

Paused ECs were assembled using the synthetic nucleic acids (Supplementary Table 3). First, a hybrid consisting of a T-DNA and its complementary RNA was formed by mixing both oligos (each 0.5 mM in 10 mM Tris/HCl, 40 mM KCl, 5 mM MgCl₂, pH 8.0) in equimolar ratio and annealing them in a PCR machine (95 °C for 2 min, 75 °C for 2 min, 45 °C for 5 min then cooling to 25 °C at 1 °C/min). RNAP (-20 μM in buffer D^RNAP) was then added at a molar ratio of 1.3:1 (T-DNA/RNA:R-NAP) and allowed to bind the hybrid for 10 min at RT. The transcription bubble was subsequently completed by adding a NT-DNA (molar ratio hybrid:NT-DNA = 1:2) and incubating the mixture for 10 min at 32 °C. Excess nucleic acids were then removed by SEC using a Superose 6 Increase 3.2/300 SEC column equilibrated in cryoEM buffer (20 mM Tris/HCl, 120 mM KOAc, 5 mM Mg(OAc)₂, 10 μM ZnCl₂, pH 7.5). Fractions (50 μl each) containing the PEC were combined and concentrated by ultrafiltration at 4 °C via an Amicon Ultra-0.5 ml unit (MWCO: 100 kDa) to a concentration of -3.3 mg/ml RNAP.

## CryoEM: sample preparation

3.8 μl of *ops*PEC was mixed with Octyl ß-D-glucopyranoside (NOG; Sigma-Aldrich) to a final concentration of 0.15% and applied to glow-discharged Quantifoil R1.2/1.3 holey carbon grids (Quantifoil Microtools GmbH, Großlöbichau) and plunged into liquid ethane using a Vitrobot Mark IV (Thermo Fisher) set at 10 °C and 100 % humidity. For assembly of *ops*PEC^Enc, 6 μM of *ops*PEC was mixed with 3x fold excess of RfaH^CC in cryoEM buffer and incubated on ice for 30 min. For opsPEC^Rec, 6 μM of *ops*PEC was mixed with 3x fold excess of RfaH^wt in cryoEM buffer supplemented with 2 mM DTT and incubated for 10 min at 37 °C. For *ops*PEC^Rec + NusA, 6 μM of *ops*PEC was mixed with 3x fold excess of RfaH^wt and 2x fold excess of NusA in cryoEM buffer supplemented with 2 mM DTT and incubated for 10 min at 37 °C. For *ops*PEC^Back, 6 μM *ops*PEC was mixed with 3x fold excess of RfaH^wt in cryoEM buffer supplemented with 2 mM DTT and incubated for 10 min

at 37 °C. NTPs were added to the mixture to a final concentration of 0.2 mM and the reaction was incubated for an additional 20 min at 37 °C. All complexes were mixed with NOG and vitrified as described for the *ops*PEC sample.

## CryoEM: Data acquisition and analysis

Data acquisition was conducted on a FEI Titan Krios G3i TEM operated at 300 kV equipped with a Falcon 3EC detector. Movies were taken for 40.57 s accumulating a total electron flux of ~40 el/Å² in counting mode at a calibrated pixel size of 0.832 Å/px distributed over 33 fractions. EPU (version 2.8.1; Thermo Fisher Scientific) was utilized for automated acquisition using a nominal defocus between −0.8 and −2 μm. A total of 9 data sets was acquired from a single grid per sample during individual sessions.

Data analysis for all datasets was conducted in a similar way within the cryoSPARC framework (version 3.2-4.0.2)[76]. To accelerate productivity, on-the-fly processing using cryoSPARC Live was conducted during data acquisition. Patch motion correction of raw movies was utilized to generate half-binned (2048 px x 2048 px) aligned micrographs. After Patch CTF estimation, the blob picker was used for initial particle picking. A box size of 192 px, fourier-cropped to 96 px was selected, 2D classification using 25 classes was applied with a circular mask diameter of 200 Å. Shiny classes were used for template-based picking using a particle diameter of 200 Å. Ab initio reconstruction of the best 2D class averages generated an initial reconstruction used for heterogeneous refinement of the whole dataset to select good particles. Reconstructions were either improved by homogeneous refinement or subsequent classifications by heterogeneous refinement or 3D variability analysis was conducted until homogeneous particle sets were determined with isotropic density distribution. Local motion correction using a box size of 384 px followed by local and global CTF refinement was applied to reconstruct cryoEM densities at full resolution by NU refinement. Local resolution estimates (blocres) were used to generate locally filtered maps for modeling.

Although all samples were vitrified in identical fashion, viewing angle distributions suggest orientation issues for some complexes (Supplementary Figs. 2, 5, 6, and 7). However, visual inspection of the reconstructions and local resolution plots did not reveal inflation by orientation bias, indicating that structure analysis was not adversely affected.

## Model building and refinement

Structures of RNAP (PDB ID 6C6S) and RfaH (PDB ID 5OND) were docked into the cryoEM map using COOT (version 0.9.8.1)[77]. Proteins and nucleic acids were manually rebuilt into the cryoEM density. The entire structure was manually adjusted residue-by-residue, supported by real-space refinement in COOT. The manually built model was refined against the cryoEM map using the real space refinement protocol in PHENIX (version 1.20_44591)[78]. The structural models were evaluated with MolProbity (version 4.5.1)[79]. Structure figures were prepared with ChimeraX (version 1.7)[80].

## Molecular Dynamics Simulations: Preparation of initial structures

The cryoEM structures of *ops*PEC bound to the autoinhibited and active states of RfaH have missing residues in several regions. In both cases, the RNAP β subunit (chain I) is missing residues 891-911 and RNAP β' subunit (chain J) is lacking residues 936-946 and 1127-1133. In the *ops*PEC bound to autoinhibited RfaH, RNAP α subunit (chain H) is missing residues 160-166 and RNAP β' subunit is lacking residues 68-92, whereas RfaH also lacks residues 102-119 and 154-162 of the inter-domain linker and KOW^α. Lastly, recruitment of RfaH to *ops*PEC in the autoinhibited state was achieved by introducing an interdomain disulfide bond through mutations F51C and S139C.

To reverse the F51C and S139C mutations in the autoinhibited state of RfaH, we used PyMOL (version 2.5.0; Schrödinger LLC). For C139S, we selected a conformation with low steric van der Waals clashes (strain = 20.82) similar to that seen in the crystal structure of autoinhibited RfaH in complex with *ops* DNA (PDB ID 5OND) after structural superimposition. In the case of C51F, the loop harboring this residue position is sufficiently displaced when compared to its conformation in the crystal structure of autoinhibited RfaH, such that the side chain sits outside the interdomain interface. Thus, we opted to choose the phenylalanine side chain rotamer with the lowest van der Waals strain (41.76).

To add missing residues, we used Modeller (version 10.4)[81] along with split regions from several templates retrieved after structural superposition. For the missing residues in RNAP β subunit (residues 891-911), we used residues 887-915 from chain I in the cryoEM structure of RfaH bound to *ops*EC (PDB ID 6C6S), extracted after structural superposition of residues 870-930 (root-mean-square deviation [RMSD] 0.48 Å against *ops*PEC bound to active RfaH; 0.42 Å against *ops*PEC bound to autoinhibited RfaH). For the autoinhibited state of RfaH, we employed the crystal structure of free full-length RfaH (PDB ID 2OUG) as template, as it contains an almost complete α-helical KOW hairpin (residues 115-156). In this case, we first split the α-helical hairpin of the template structure in its two helices, and superimposed residues 116-124 (RMSD 0.67 Å) and residues 150–156 (RMSD 0.19 Å) against the corresponding residues in the structure of *ops*PEC bound to autoinhibited RfaH. Finally, for regions in RNAP subunits uniquely missing in *ops*PEC bound to autoinhibited RfaH, we used residues 154-168 of the α subunit (chain J) and residues 61-97 of the β' subunit (chain H) of *ops*PEC bound to active RfaH as templates. All other missing regions were subjected to loop modeling. During the modeling procedure, non-missing residues coming from the cryoEM *ops*PEC were not allowed to move.

Given the larger number of missing residues in *ops*PEC bound to autoinhibited RfaH, the good structural superposition of the *ops*PEC bound to either RfaH state (RMSD 0.75 Å), and that our interest was on simulating the fold-switch of RfaH from the recruited into the active state, we continued our work using only the structure of *ops*PEC from the active state for both RfaH:*ops*PEC complexes. To do this, we replaced the *ops*PEC from the cryoEM structure of the autoinhibited RfaH:*ops*PEC complex with the one from the active RfaH complex after superposition of the residues around 6 Å of RfaH (RMSD 1.3 Å).

Lastly, the completed structures of both *ops*PEC:RfaH complexes were subjected to an energy minimization in explicit solvent on GROMACS (version 4.5)[82]. In this process, each complex was parameterized using the AMBER99SB-ILDN[83] force field, placed in a cubic box with 1.0 nm of padding filled with SPC/E water molecules[84] and neutralized with counter ions (i.e. sodium) before energy minimization using the steepest descent method until reaching a maximum total force <1000 kJ/mol/nm (Supplementary Table 4).

### Molecular Dynamics Simulations: All-atom dual-basin structure-based models

The energy-minimized *ops*PEC:RfaH complexes were extracted from each system and used for the subsequent generation of all-atom structure-based models (SBM)[85,86] using SMOG 2 (version 2.4.5)[87]. In these models, all native contacts are given attractive single-basin Gaussian interactions, whereas non-native interactions are given repulsive terms to ensure that atoms have a defined excluded volume[85]. Native interactions are defined as all atom-atom pairs at a maximum contact distance 6 Å, with a "shadowing" screening parameter radius for occluded contacts of 1 Å, and a sequence separation of at least 3 residues for protein-protein contacts whereas no sequence separation is imposed to nucleic acid contacts. The complete functional form of these potentials is described elsewhere[85].

These all-atom SBM were further utilized to generate a dual-basin model of *ops*PEC bound to RfaH that has both its autoinhibited and active states as explicit minima. These dual-basin models were created by combining all native contacts and dihedrals from both the autoinhibited and active states of RfaH. Atom-atom pairs participating in native interactions in both conformations and having a difference in distance >10% (246 contacts) were given a dual-basin Gaussian contact potential[88]. Likewise, dihedrals that differ between both RfaH conformations by >20% (940 dihedrals) were given dual-basin dihedral potentials[44]. Given the interest in simulating the transition from the autoinhibited to the active state of RfaH, parameters for bonds and angles in these dual-basin SBM were kept from the energy-minimized structure of *ops*PEC bound to the active state of RfaH.

A total of 500 independent simulations using Langevin dynamics of the fold-switch of RfaH from the autoinhibited to the active state were performed in the absence and presence of *ops*PEC (Supplementary Table 4). The dual-basin SBM simulations are run in reduced units, where the length scale, time scale, mass scale, and energy scale are all 1[87]. Simulations were run using a time step of 0.002 reduced units (τ) and a temperature of 0.67 reduced units, collecting data every 1,000 timesteps (2 τ) and with each simulation continued until the RMSD of RfaH against the cryoEM structure of the autoinhibited state reached values below 5 Å (maximum trajectory length = 11,600 τ; Gaussian distribution mean trajectory length = 2200 τ). Based on the comparison of SBMs with explicit solvent simulations, it is estimated that 1 τ is equivalent to 1 ns[89]. The choice of temperature was made based on preliminary simulations in which the DNA:DNA and DNA:RNA interactions started to break apart at 0.75 reduced units.

All 500 simulations were analyzed throughout this work. We also split the data into two halves and subjected each half to re-analysis, demonstrating that the results presented in this work do not differ significantly when only the first half, second half, or the whole simulated data is used for analysis (Supplementary Fig. 8).

### Structural superpositions and calculation of swiveling angles

For analysis of swiveling angles between two ECs, RNAP molecules were superimposed on the core region (consisting of α-I, α-II, β: 1-30, 140-150, 445-455, 513-832, 1056-1240, β': 343-368, 421-786, and ω) in PyMOL. The angle between the two swivel modules (defined as regions: β: 1241-1341, β': 16-347, 369-420, 787-931, 946-1126, 1135-1373) was then calculated using the angle_between_domains script (available from https://raw.githubusercontent.com/speleo3/pymol-psico/master/psico/orientation.py).

### Calculation of hybrid diameters and upstream/downstream duplex DNA angles

To calculate the hybrid radius and angle between the upstream and downstream duplex DNAs within an EC/PEC, the nucleic acids were extracted in PyMOL, split into individual upstream DNA, downstream DNA and RNA:DNA hybrid models, and then analyzed using the online 3DNA 2.0 tool[90]. This analysis yields the hybrid diameter as well as the helix vectors defining the two duplex DNAs. The helix vectors were then translated to the same origin and the angle between them was then calculated using the "get_angle" command within PyMOL.

### Reporting summary

Further information on research design is available in the Nature Portfolio Reporting Summary linked to this article.

## Data availability

CryoEM reconstructions have been deposited in the Electron Microscopy Data Bank (https://www.ebi.ac.uk/pdbe/emdb) under accession codes EMD-17626 (*ops*PEC), EMD-17657 (nc-*ops*PEC), EMD-17679 (*ops*PEC^Enc), EMD-17646 (nc-*ops*PEC^Enc), EMD-17668 (*ops*PEC^Rec state 1), EMD-17632 (*ops*PEC^Rec state 2), EMD-17686 (nc-*ops*PEC^Rec state 1), EMD-

17647 (nc-*ops*PEC^Rec state 2), EMD-17681 (*ops*PEC^back), and EMD-17685 (*ops*PEC^Rec,NusA). Structure coordinates have been deposited in the RCSB Protein Data Bank (https://www.rcsb.org) with accession codes 8PDY (*ops*PEC), 8PH9 (nc-*ops*PEC), 8PIB (*ops*PEC^Enc), 8PFG (nc-*ops*PEC^Enc), 8PHK (*ops*PEC^Rec state 1), 8PEN (*ops*PEC^Rec state 2), 8PIM (nc-*ops*PEC^Rec state 1), 8PFJ (nc-*ops*PEC^Rec state 2), 8PID (*ops*PEC^back), and 8PIL (*ops*PEC^Rec,NusA). All-atom structure-based models of *ops*PEC^Enc and *ops*PEC^Rec, representative structures for intermediates retrieved from 500 simulations of RfaH refolding in the absence and presence of *ops*PEC, and representative trajectories of RfaH fold-switching while bound to *ops*PEC, are available at Zenodo [https://doi.org/10.5281/zenodo.10493315]. All other data are contained in the manuscript or the Supplementary Information. Structure coordinates used in this study are available from the RCSB Protein Data Bank (https://www.rcsb.org) under accession codes 2OUG, 5OND, 5VOI, 6ALF, 6C6S, 6RH3, 7YPA, 8EG7, 8EG8, 8EH8, and 8FVW. Source data are provided with this paper.

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

## Acknowledgements

This work was supported by the Deutsche Forschungsgemeinschaft (WA 1126/11-1, project number 433623608, to M.C.W.; INST 130/1064-1 FUGG to Freie Universität Berlin; RO 617/21-1 to P. Rösch), National Institutes of Health (GM067153 to I.A.), the National Agency for Research and Development (ANID) through Millennium Science Initiative Program (ICN17_022), Fondo de Desarrollo Científico y Tecnológico (FONDECYT 1201684 to C.A.R.-S) and the Academy of Finland (341962 to G.A.B.). J.G.-H. was supported by an ANID Doctoral Scholarship (PFCHA 21212113). We thank Kurt Fredrick for the gift of 70 S ribosomes, Simon Hofer and Marinus Thein for initial work on RfaH<sup>CC</sup>, and Dmitri Svetlov for discussions and editing.

## Author contributions

P.K.Z purified and characterized proteins and prepared complexes for CryoEM and prepared figures. N.S. prepared complexes for CryoEM. B.W. purified mutant RNAP variants and performed in vitro pause assays. T.H. performed cryoEM experiments. P.K.Z. and S.H.K. carried out model building and refinement of the CryoEM structures, assisted by B.L. J. G.-H. and C.A.R.-S. performed MD simulations. All authors contributed to the data analysis. I.A. and S.H.K. wrote the first draft and revised the manuscript with contributions from C.A.R.-S., G.A.B. and M.C.W. All authors participated in preparation of the final manuscript.

## Funding

## Competing interests

The authors declare no competing interests.
