## [Peer Review File · Nature Communications]

Concerted transformation of a hyper-paused transcription complex and its reinforcing proteinREVIEWER COMMENTS

Reviewer #1 (Remarks to the Author):

Function of transcribing RNA polymerase (RNAP) is influenced by interacting with transcription elongation factor and also DNA and RNA secondary structures. In this study, authors investigated well-characterized transcription pausing of *E. coli* RNAP at ops site and how it is influenced by NusG-type transcription elongation factor RfaH. Dr. Artsimovitch, one of corresponding authors, published a paper in Kang et al (Cell, 2018) with the Darst group for reporting the cryo-EM structure of the elongation complex containing ops and RfaH, which documented the formation of opt non-template DNA hairpin as well as its interaction with RfaH, but didn't provide any mechanistic insight into the transcription pausing by opt and/or RfaH. In this current study, authors claimed that a previous study used DNA/RNA scaffold, which was not representing transcription pausing at opt site (lacking DNA complementarity between template and non-template DNA, and short upstream DNA duplex). Thus, in this study, authors modified DNA/RNA scaffold to form the opt pausing complex with and without RfaH, to reveal the mechanism of transcription pausing, RfaH loading and its conformational change, and pause escape by solving a series of cryo-EM structures, structure-based mutagenesis, and molecular dynamic simulation.

Authors revealed from this study compare with a previous study published in Kang et al (Cell, 2018) that the nt-DNA with ops site forms a short DNA hairpin, which is recognized by RNAP and induces conformational change (swiveled state) as well as making EC inactive for RNA extension by stabilizing pre-translocated state. In addition, the opt hairpin recruits RfaH followed by its conformational change from inactive to active forms by release its C-terminal domain studied by cryo-EM and MD simulation.

Questions:

In Figure 7, it was not clearly explained how EC complex (e) is formed from the opsPEC. There is no arrow indicating its formation from opsPEC.

optPEC maintains pre-translocation state, which blocks NTP binding at the active site, as well as preventing template DNA and incoming NTP base pairing. Authors discuss a possible mechanism of NTP binding and incorporation from the optPEC without DNA/RNA translocation (or address it by solving the cryo-EM structure of optPEC with non-hydrolysable GTP, or 3'-deoxy RNA plus GTP combination).

Authors should discuss about how NusG is released from the EC when it reaches opt pause site. In Kang et al (Cell 2018) study, NusG on EC prevents formation of the opt hairpin (ntDNA in the transcription

bubble was disordered). NusG associates majority of elongating RNAP in E. coli and if so, how EC at opt site can form ntDNA hairpin prior to RfaH association?

In supplemental movies showing RfaH conformational change (MD simulation), some frames show bad steric clash between RfaH (a linker and CTD) and core enzyme. Did authors use force field allows such steric clash?

In cryo-EM data processing, some structures (opsPEC or nc-opsPEC, Extended Data Fig. 2 d and e) show particle orientation issue on particle angular distributions plots but some are not (Ext Data Fig. 6 b and d). Why? Did authors use detergent only for some complexes?

Reviewer #2 (Remarks to the Author):

The well-written manuscript by Zuber et al provides a structural model for how RfaH initially interacts with paused RNAP, how RfaH and its RNA target undergo structural changes following the initial encounter interaction, and how the original pause becomes a long-lived pause via backtracking. Biochemical studies demonstrate how GreA and Mfd are capable of reversing the backtracked complex into an elongation competent form. In my opinion this work is thorough, expertly executed, and will be of high interest to those interested in transcription. I also note that the methods are highly detailed and thorough. Additional minor points are listed below.

1. In the introduction you might want to mention a recent paper demonstrating that E. coli NusG suppresses pausing genome wide (DOI: 10.1073/pnas.2221114120).

2. Line 71-72. Extended data Fig. 1 and 2 seem to be mixed up in their order.

3. Lines 73-74. ...NGN is accommodated on RNAP...

4. Line 80. ...expression by several hundred fold.

5. Line 99. Consider replacing "merely" with "short"

6. Line 195. I think this reference should be to Fig. 1b.
7. Line 264. The figure uses "structure 1 and 2" rather than "complex 1 and 2"
8. Line 297. Figure 5...
9. Extended data Fig. 7 (line 339) is mentioned before Extended data Fig. 6 (line 356).
10. Lines 381-382. I don't understand how you came to this conclusion based on what is described above. Please provide more explanation.
11. Line 452. Perhaps you should say ...to be determined whether the 30S or 70S ribosome is...
12. Line 579. You already mentioned TRIS in several places. Move the full name of TRIS to the first time I used or don't use at all.
13. Line 596. 70S ribosomes, not just 70S.
14. Line 621. Are these in-house programs available to the public? This sounds like original code (see line 803).
15. Line 640. What is U*A?
16. The use of yellow on a white background in figures is hard to see. I encourage you to replace all yellow with another color that is darker.
17. Extended Discussion. It is not correct to say that "The common feature of all PECs is failure to translocate..." NusG-dependent pauses are primarily in the post-translocated state.
18. Extended Data Figure 2. Panels a and B are swapped in the figure and legend.

19. Extended data Figure 4 legend. In one case you have KOW α rather than KOW α .

20. Extended data Figure 7e legend. Delete "within".

Reviewer #3 (Remarks to the Author):

The manuscript “Concerted transformation of a hyper-paused transcription complex and its reinforcing protein” by Zuber et al. presents a systematic structural study explaining the role of a NusG-like factor RfaH on bacterial RNA polymerase transcription. RfaH binds to an ops sequence in a paused RNAP elongation complex and undergoes conformational changes. The current study reports a series of related structures to provide snapshots of ops-pausing, RfaH-binding, domain flipping, and structural rearrangement of the C-terminal KOW domain. This comprehensive study provides an insight into RfaH role in stabilizing the paused RNAP elongation complex potentially for engaging ribosome for specific protein synthesis.

Major comments

The section “Refolding landscape of RfaH upon recruitment to opsPEC” which appears to be a major claim of the study, is based on an MD simulation calculation. I assume the simulation is to show the transition between the KOW (closed) and KOW (open) forms. Then the following sections discuss KOW's role in engaging ribosome. It is unclear how this study enhances our understanding of KOW's roles. The authors may need to clearly show experimental evidences for their claims.

Authors should clearly show, at least in the extended data section, what each structure represents in the context of the results and discussion in the manuscript.

While the experimental density for overall RNA polymerase is expected to be good, the density maps sections for the discussed region appear to be of low resolution. The authors should show zoomed local resolution maps for the region including the NT hairpin, interacting RNAP, and RfaH in extended data. If the resolution of the region is lower, then explain how the discussed interactions are modeled.

The authors may need to establish the importance of the interactions shown in Fig. 3 by site-directed mutagenesis studies.

The MD simulation analysis appears to be overemphasized. Fig. 5 may be moved to the Extended Data section.

Other comments

Page 3. The sentence “NusG paralogs function alongside NusG and have just a few targets, which are essential in some conditions, e.g., during infection” is unclear. Please expand.

Page 8. How do the authors conclude the half-life of 8 seconds in the statement “The wt RNAP paused at U11 with a half-life of 8 seconds, whereas β R542A substitution delayed escape \sim 2.5 fold (Fig. 2h)”? Figure 2h are structural states not a time plot.

Page 17. The hairpin is described as having a “unique geometry”. Can authors define this?

In PDB validation reports, Section 7.2 estimates the mass of RNAP \sim 130kDa at 0.495 (recommended) contour level. The actual mass of RNAP is significantly high. Also, at 0.497 contour level, the map covers only \sim 75% of atoms. The contour level may be corrected.

Response to Reviewer Comments

Reviewer comments are repeated in regular font, responses are in red, changed text passages are highlighted in yellow. Line numbers in responses refer to line numbers of the revised manuscript.

Coordinate and map files are available at: <https://box.fu-berlin.de/s/57XCSJPSH8RPeKT>

Reviewer #1 (Remarks to the Author):

Function of transcribing RNA polymerase (RNAP) is influenced by interacting with transcription elongation factor and also DNA and RNA secondary structures. In this study, authors investigated well-characterized transcription pausing of *E. coli* RNAP at ops site and how it is influenced by NusG-type transcription elongation factor RfaH. Dr. Artsimovitch, one of corresponding authors, published a paper in Kang et al (Cell, 2018) with the Darst group for reporting the cryo-EM structure of the elongation complex containing ops and RfaH, which documented the formation of opt non-template DNA hairpin as well as its interaction with RfaH, but didn't provide any mechanistic insight into the transcription pausing by opt and/or RfaH. In this current study, authors claimed that a previous study used DNA/RNA scaffold, which was not representing transcription pausing at opt site (lacking DNA complementarity between template and non-template DNA, and short upstream DNA duplex). Thus, in this study, authors modified DNA/RNA scaffold to form the opt pausing complex with and without RfaH, to reveal the mechanism of transcription pausing, RfaH loading and its conformational change, and pause escape by solving a series of cryo-EM structures, structure-based mutagenesis, and molecular dynamic simulation.

Authors revealed from this study compare with a previous study published in Kang et al (Cell, 2018) that the nt-DNA with ops site forms a short DNA hairpin, which is recognized by RNAP and induces conformational change (swiveled state) as well as making EC inactive for RNA extension by stabilizing pre-translocated state. In addition, the opt hairpin recruits RfaH followed by its conformational change from inactive to active forms by release its C-terminal domain studied by cryo-EM and MD simulation.

Questions:

In Figure 7, it was not clearly explained how EC complex (e) is formed from the opsPEC. There is no arrow indicating its formation from opsPEC.

Sorry for this not being clear enough. We now added a forward arrow with labeling, indicating that (e) is formed from (d) (*opsPEC*^{REC}) by elongation followed by backtracking. Please also see following point. We also want to point out that in the original Fig. 7 we focused on the RfaH-centric pathway and ignored pause- and RfaH-bypass mechanisms. We now also indicate these possibilities in the modified Fig. 7:

optPEC maintains pre-translocation state, which blocks NTP binding at the active site, as well as preventing template DNA and incoming NTP base pairing. Authors discuss a possible mechanism of NTP binding and incorporation from the optPEC without DNA/RNA translocation (or address it by solving the cryo-EM structure of optPEC with non-hydrolysable GTP, or 3'-deoxy RNA plus GTP combination).

We have investigated how *opsPEC*^{Rec} state resumes transcript elongation by determining the structure of the complex after adding NTPs. The outcome is described in the section entitled "The *opsPEC*^{Rec} can be arrested". We show that the *opsPEC*^{Rec} can add nucleotides but backtracks to the *opsP* site following the incorporation of two or more nucleotides. The observation of the *opsPEC*^{Rec} with the extended transcript indicates that the forward translocation takes place transiently, allowing NTP binding and incorporation. In other words, a small fraction of the post-translocated state must be present in the *opsPEC*^{Rec} preparation, but such fraction cannot be reliably detected in the background of the pre-translocated state. Indeed, at the typical resolution of cryo-EM structures, the difference between the post- and pre-translocated states may be merely the absence of the density for the RNA 3' terminal nucleotide in the former. The mixture of the dominant pre- and minor post-translocated states

then differs from the pure pre-translocated state by a slightly reduced density of the 3' terminal RNA nucleotide. Noteworthy, a recent study of pause sites from Landick and Darst labs (PMID: 36795753) also failed to report structures of post-translocated states, despite the biochemical analysis suggesting the presence of a detectable fraction of the post-translocated states at investigated pause sites. It thus appears that the detection of a small fraction of the post-translocated state at predominantly pre-translocated pause sites is currently beyond the capabilities of the cryo-EM analysis. We further argue that our experiment with the NTP-supplemented *opsPEC^{Rec}* is more informative than an attempt to bias 3'-deoxy EC forward by GTP. 3'OH forms important interactions stabilizing either pre- (to N458, Q929) or post- (to Mg1) translocated states, so using 3'-deoxy RNA may lead to a number of artifacts. At the same time, GMPCPP is relatively weak in biasing EC forward. Considering that we only observed backtracked *opsPEC^{Rec}* after the NTP addition, there is little chance that GMPCPP would be able to bias the *opsPEC^{Rec}* forward to a measurable extent. It appears that *opsPEC^{Rec}* creates such a deep depression in the potential energy landscape that any closely located downstream complex slides back to the pre-translocated *opsP*.

To clarify this point, we explicitly stated that *opsPEC^{Rec}* can transiently adopt a post-translocated state (line 352):

These findings establish that *opsPEC^{Rec}* is elongation-competent (i.e., can transiently adopt a post-translocated state) and that, following nucleotide addition, RNAP slides back, generating *opsPEC^{Back}*.

Authors should discuss about how NusG is released from the EC when it reaches opt pause site. In Kang et al (Cell 2018) study, NusG on EC prevents formation of the opt hairpin (ntDNA in the transcription bubble was disordered). NusG associates majority of elongating RNAP in *E. coli* and if so, how EC at opt site can form ntDNA hairpin prior to RfaH association?

Transcription factors dynamically interact with RNAP, and in this respect are distinct from RNAP subunits. Our earlier research indicates that NusG dissociates from EC at a rate of approximately 1 s^{-1} at 25 °C (Fig. 6 in <https://doi.org/10.7554/elife.18096>). This rate implies that transcriptional pausing at *ops* allows enough time for NusG to dissociate, enabling the formation of the NT-DNA hairpin. This hairpin then facilitates recruitment of RfaH and disfavors NusG rebinding. Given that the formation of the NT-DNA hairpin is also a reversible process, high concentration of EC, NusG and long incubation times may allow NusG rebinding to *opsPEC*. However, NusG rebinding is unlikely to happen in the presence of RfaH, which easily outcompetes NusG at *ops*, as we have previously shown happens *in vitro* and in the cell (<https://doi.org/10.1038/emboj.2008.268>). Furthermore, *in vivo* analysis of NusG association with RNAP (PMID: 19150431) demonstrated that NusG is recruited to the EC slowly, and thus would be unlikely to bind at *ops* sites, which are typically located within first 100 nts of the mRNA.

We now cite these results in the revised manuscript (line 437):

E. coli NusG makes no contacts to DNA¹³ and readily dissociates from the EC⁶⁴. NusG is excluded from the EC by RfaH^{13,29} and, conceivably, by NT-DNA structures.

In supplemental movies showing RfaH conformational change (MD simulation), some frames show bad steric clash between RfaH (a liner and CTD) and core enzyme. Did authors use force field allows such steric clash?

The steric clashes seen in the supplemental movies might be an artifact of the smoothing and visualization utilized for presentation purposes. We uploaded representative trajectories of the

refolding simulations matching the two routes showcased in Supplementary Movies 1 (route 1) and 2 (route 2) to a Zenodo repository (<https://doi.org/10.5281/zenodo.10493315>).

Regarding the features of the force field, the Gaussian potential for native atom-atom contacts is described by an attractive Gaussian minimum centered at the distance of the atom-atom pairs in the native structure and a repulsive wall, and all atom pairs not in contact in the native state are also given repulsive interactions. The functional form of the repulsive part of the all atom interactions is:

$$\varepsilon_{NC} \left(\frac{\sigma_{NC}}{r_{ij}} \right)^{12}$$

Where the strength of the potential, ε_{NC} , is 0.1, and the distance for non-native atom-atom interactions, σ_{NC} , is = 0.25 nm. Finally, r_{ij} is the distance between the atom pairs during the simulation.

In cryo-EM data processing, some structures (opsPEC or nc-opsPEC, Extended Data Fig. 2 d and e) show particle orientation issue on particle angular distributions plots but some are not (Ext Data Fig. 6 b and d). Why? Did authors use detergent only for some complexes?

Indeed, viewing angle distributions suggest orientation issues for some complexes. However, visual inspection of the maps as well as local resolution plots did not show inflation by orientation bias. Regarding sample preparation, all samples were treated in identical fashion, *i.e.* by addition of n-octylglucoside to a final concentration of 0.15 % (w/v) immediately before vitrification. We therefore assume that the differences in viewing angle distributions are most likely due to chance differences in ice thickness rather than due to differences in sample preparation or complex composition. As we did not discern adverse effects on the reconstructions, we refrained from repeating cryoEM data collection from newly prepared specimen.

We added a corresponding statement to the revised Methods section (line 739):

Although all samples were vitrified in identical fashion, viewing angle distributions suggest orientation issues for some complexes (Extended Data Figs. 2, 5, 6, and 7). However, visual inspection of the reconstructions and local resolution plots did not reveal inflation by orientation bias, indicating that structure analysis was not adversely affected.

Reviewer #2 (Remarks to the Author):

The well-written manuscript by Zuber et al provides a structural model for how RfaH initially interacts with paused RNAP, how RfaH and its RNA target undergo structural changes following the initial encounter interaction, and how the original pause becomes a long-lived pause via backtracking. Biochemical studies demonstrate how GreA and Mfd are capable of reversing the backtracked complex into an elongation competent form. In my opinion this work is thorough, expertly executed, and will be of high interest to those interested in transcription. I also note that the methods are highly detailed and thorough. Additional minor points are listed below.

1. In the introduction you might want to mention a recent paper demonstrating that *E. coli* NusG suppresses pausing genome wide (DOI: 10.1073/pnas.2221114120).

Done

2. Line 71-72. Extended data Fig. 1 and 2 seem to be mixed up in their order.

Fixed; Extended Data Fig. 1 was referenced in error

3. Lines 73-74. ...NGN is accommodated on RNAP...

Done

4. Line 80. ...expression by several hundred fold.

Done

5. Line 99. Consider replacing "merely" with "short"

Replaced

6. Line 195. I think this reference should be to Fig. 1b.

No, it is 1a because 1b are the scaffolds. We drew a box around panel b to make this easier to see.

7. Line 264. The figure uses "structure 1 and 2" rather than "complex 1 and 2"

We changed "structures" to "states" in the figure and in the text.

8. Line 297. Figure 5...

Fixed

9. Extended data Fig. 7 (line 339) is mentioned before Extended data Fig. 6 (line 356).

Fixed; this should have been Extended Data Fig. 6

10. Lines 381-382. I don't understand how you came to this conclusion based on what is described above. Please provide more explanation.

To better explain, we revised the text to say (line 394):

NusA can aid in coupling transcription to translation²¹ and our data show that NusA can interact with RNAP and the ribosome in the RfaH-modified EC. Thus, NusA could cooperate with the RfaH KOW domain to provide additional docking sites the pioneering ribosome.

11. Line 452. Perhaps you should say ...to be determined whether the 30S or 70S ribosome is...

Done

12. Line 579. You already mentioned TRIS in several places. Move the full name of TRIS to the first time I used or don't use at all.

Thanks, we removed the full name; Tris is a common name anyway.

13. Line 596. 70S ribosomes, not just 70S.

Done

14. Line 621. Are these in-house programs available to the public? This sounds like original code (see line 803).

Indeed, for historical reasons in-house programs have been used, but publicly available programs that serve the same purposes are available. We now mention this in the revised text (line 653):

Data were processed using in-house programs that yield the same information as publicly available programs, such as NMRRIpe⁷⁶.

15. Line 640. What is U*A?

A lab name for this starting NTP subset; deleted

16. The use of yellow on a white background in figures is hard to see. I encourage you to replace all yellow with another color that is darker.

We believe that yellow ops site in structure figures is actually quite well visible and would like to stick with it, as it is well contrasted with other structural features. We agree that in schemes, yellow ops was somewhat difficult to see. We therefore added a narrow black outline in each case.

17. Extended Discussion. It is not correct to say that "The common feature of all PECs is failure to translocate..." NusG-dependent pauses are primarily in the post-translocated state.

We thank the reviewer for pointing out this discrepancy. We modified the text to limit the generalization to the pauses that we discuss. The corresponding text of Extended Discussion now reads as follows:

The common feature of classic factor-independent PECs is failure to translocate and load the acceptor T-DNA base into the active site. *ops*PEC and *con-e*PEC are pre-translocated, *his-e*PEC equilibrates between pre- and half-translocated states (only RNA translocates), and *his*PEC is half-translocated. The asynchronous (half) translocation leads to tilting of the nucleobases of the RNA:DNA hybrid relative to the helical axis. Since incomplete translocation is characteristic for all these pauses, the DFJ and UFJ are likely mainly responsible for the translocation block.

We also included a reference to the consensus PEC in the main text, as follows (line 124):

Consistently, while in structures of *his*PECs the RNA:DNA hybrid adopted a half-translocated state^{37,38} (RNA post-translocated, DNA pre-translocated), our *ops*PEC resides in the pre-translocated state (Fig. 2a), as was observed for consensus PEC³⁹.

18. Extended Data Figure 2. Panels a and B are swapped in the figure and legend.

Fixed

19. Extended data Figure 4 legend. In one case you have KOWa rather than KOWalpha.

Fixed

20. Extended data Figure 7e legend. Delete "within".

Done

Reviewer #3 (Remarks to the Author):

The manuscript “Concerted transformation of a hyper-paused transcription complex and its reinforcing protein” by Zuber et al. presents a systematic structural study explaining the role of a NusG-like factor RfaH on bacterial RNA polymerase transcription. RfaH binds to an ops sequence in a paused RNAP elongation complex and undergoes conformational changes. The current study reports a series of related structures to provide snapshots of ops-pausing, RfaH-binding, domain flipping, and structural rearrangement of the C-terminal KOW domain. This comprehensive study provides an insight into RfaH role in stabilizing the paused RNAP elongation complex potentially for engaging ribosome for specific protein synthesis.

Major comments

The section “Refolding landscape of RfaH upon recruitment to opsPEC” which appears to be a major claim of the study, is based on an MD simulation calculation. I assume the simulation is to show the transition between the KOW (closed) and KOW (open) forms. Then the following sections discuss KOW's role in engaging ribosome. It is unclear how this study enhances our understanding of KOW's roles. The authors may need to clearly show experimental evidences for their claims.

Our MD analysis did not aim at “enhancing our understanding of KOW's roles”, which are well known and published. The KOW's roles in RfaH autoinhibition and transcription-translation coupling hinge on its ability to refold. An open question that we sought to answer was whether the refolding process is influenced by the opsPEC context. The decision of performing MD simulation to follow the fold-switch of RfaH upon binding to opsPEC was made because our cryoEM structures provide structural snapshots immediately before initiation and after completion of the refolding process, and therefore the ideal experimental approach to address this question is MD.

In this regard, it is worth noting that the type of simulations utilized herein, which consist of the generation of structure-based models (SBMs) that reduce the ruggedness of classic MD force fields by accounting for distance-based contacts between atoms in proximity in the native state instead of utilizing empirical force fields, have been extensively validated to provide accurate depictions of other refolding processes. Exemplar cases are the examination of accommodation of tRNA into the ribosome, with reversible fluctuations and motions observed in simulations using SBM for accommodation in different conformations being validated by particularly challenging single-molecule FRET experiments

(<https://doi.org/10.1261/rna.2035410>). Likewise, the gate-opening mechanism of TcB toxin, which requires refolding of a beta propeller, was also explored using SBMs (<https://doi.org/10.1038/s41586-018-0556-6>). Finally, direct experimental observations of the conformational dynamics of influenza hemagglutinin upon changes in pH that mimic the transition between pre- to post-fusion states (<https://doi.org/10.1016/j.cell.2018.05.050>) were consistent with SBM applied to simulate the large-scale conformational rearrangement of this protein trimer (<https://doi.org/10.1073/pnas.1412849111>).

In contrast to the proteins mentioned above, which are several hundreds of kilodaltons in size and with conformational changes that imply significant changes in distance that are discernible via FRET, the design of experiments to pursue the direct visualization of the conformational changes during the refolding of RfaH KOW, a domain of merely 50 residues, upon binding to opsPEC would be significantly more challenging. Based on this argument, on the several studies using SBMs that demonstrate their utility to explore conformational changes that are consistent with further experimental analysis, and on the fact that the direct observation of these structural changes experimentally would have an impact and merit on its own, we refrain from pursuing these experiments in the current study.

Authors should clearly show, at least in the extended data section, what each structure represents in the context of the results and discussion in the manuscript.

We are unsure what the reviewer meant, as we did show these structures in Figure 7. We added a brief summary of the main structures that delineate the RfaH pathway at the beginning of the Discussion (line 400):

Early findings that RfaH binding to the transcribing RNAP requires sequence-specific contacts to the NT DNA³² and dramatic structural rearrangements^{12,34} prompted us to propose a model for co-transcriptional recruitment and activation of RfaH. Here, we report structural and in silico data that support and extend this model (Fig. 7). First, we present a structure of *E. coli* RNAP paused at *ops*, an archetypal regulatory site that recruits RfaH. Most notably, the structure reveals an 8-nt-long NT DNA hairpin (*ops*H_P) that triggers RNAP swiveling to stabilize a catalytically inactive pre-translocated state while displaying multiple recognition motifs for RfaH. Second, we present a structure of a highly transient, yet functionally crucial, encounter complex that captures autoinhibited RfaH bound to *ops*PEC. In this structure, still autoinhibited RfaH binds to, and repositions, the *ops*H_P to initiate expansion of the transcription bubble and the RNA:DNA hybrid, stabilizing the paused state. Third, we present a structure of RfaH fully engaged with RNAP in an even deeper paused state, which is likely necessary to recruit the pioneering ribosome. Fourth, we present a structure that captures an unsuccessful attempt of RfaH-bound RNAP to escape the recruitment site, leading to a backtracked state which is rescued by accessory factors.

Alternatively, we could put this description into the Extended Data, but the Discussion is arguably the best place for this summary.

While the experimental density for overall RNA polymerase is expected to be good, the density maps sections for the discussed region appear to be of low resolution. The authors should show zoomed local resolution maps for the region including the NT hairpin, interacting RNAP, and RfaH in extended data. If the resolution of the region is lower, then explain how the discussed interactions are modeled.

We now provide zoomed local resolution maps for each complex, covering relevant elements (such as *ops* hairpin, RfaH or NusA if contained; added to Extended Data Figs. 2, 5, 6 and 7). We carefully checked which structural features we describe in the text. All of these features can be reliably deduced from the reconstructions at the respective local resolution levels.

The authors may need to establish the importance of the interactions shown in Fig. 3 by site-directed mutagenesis studies.

We are unsure which interactions the reviewer is referring to. RfaH is among the most heavily mutagenized proteins. We did carry out exhaustive mutagenesis of RfaH interactions with RNAP and the *ops*H_P in studies that preceded structural analysis of RfaH-EC complexes (doi: [10.1111/j.1365-2958.2010.07056.x](https://doi.org/10.1111/j.1365-2958.2010.07056.x); doi: [10.1016/j.molcel.2007.02.021](https://doi.org/10.1016/j.molcel.2007.02.021)). For example, we substituted all but one RfaH residue (Val) that we modeled to interact with the NT DNA, which we did not expect to form a hairpin at that time, for Ala. The effects of these substitutions were fully consistent with contacts observed in Kang et al. 2018, as shown in Fig. 3 of [10.1016/j.cell.2018.05.017](https://doi.org/10.1016/j.cell.2018.05.017) and in this study.

The MD simulation analysis appears to be overemphasized. Fig. 5 may be moved to the Extended Data section.

While we acknowledge the reviewer's comment, we wish to point out that the MD simulations in Fig. 5 have a twofold significance. First, these *in silico* experiments demonstrate that refolding of the KOW domain does not require the *ops*PEC-KOW interactions to occur, and that the role of the *ops*PEC is thus limited to triggering the dissociation of the NGN and KOW domains. Second, wet lab experiments to demonstrate the steps in the refolding of RfaH upon binding to *ops*PEC are particularly challenging given the small size of the KOW domain. Several previous studies have attempted to describe the refolding pathways of RfaH KOW either using the isolated domain or the full-length protein, but the cryoEM structures presented in this work along with the MD simulations performed using SBMs provided the opportunity to explore for the first time how the refolding of RfaH KOW occurs upon binding to *ops*PEC. Thus, we believe that these data belong to the main body of the manuscript.

Other comments

Page 3. The sentence "NusG paralogs function alongside NusG and have just a few targets, which are essential in some conditions, e.g., during infection" is unclear. Please expand.

We thank the reviewer for pointing out that our use of "targets" could be confusing. We instead now wrote (line 56)

NusG paralogs function alongside NusG to control expression of just a few genes which are essential only under some conditions, e.g., during infection²⁴.

Page 8. How do the authors conclude the half-life of 8 seconds in the statement "The wt RNAP paused at U11 with a half-life of 8 seconds, whereas β R542A substitution delayed escape ~2.5 fold (Fig. 2h)"? Figure 2h are structural states not a time plot.

There must be some confusion – in our manuscript version, Fig. 2h shows a denaturing gel with single-round pause assays.

Page 17. The hairpin is described as having a "unique geometry". Can authors define this?

We do not think that the hairpin has a unique geometry; rather, we propose that similar hairpins may form on the surface of RNAP during elongation. Here, we described the *ops*PEC geometry as unique, with "The *ops*PEC structure reveals a unique geometry, a canonical 10-bp hybrid

but 11-nt long ss NT DNA that forms the hairpin stabilized by a multitude of positively-charged RNAP residues (Fig. 2d).”

In PDB validation reports, Section 7.2 estimates the mass of RNAP ~130 kDa at 0.495 (recommended) contour level. The actual mass of RNAP is significantly high. Also, at 0.497 contour level, the map covers only ~75 % of atoms. The contour level may be corrected.

We thank the reviewer for catching this mistake. We corrected the recommended contour levels such that maps cover the models almost entirely:

Complex	Recommended contour level
opsPEC	0.25
nc-opsPEC	0.32
opsPEC ^{Enc}	0.24
nc-opsPEC ^{Enc}	0.25
opsPEC ^{Rec} state 1	0.24
opsPEC ^{Rec} state 2	0.25
nc-opsPEC ^{Rec} state 1	0.25
nc-opsPEC ^{Rec} state 2	0.25
opsPEC ^{back}	0.24
opsPEC ^{Rec,NusA}	0.25

We are in the process of updating the PDB submissions. The PDB is presently slow to respond and process updates, and no other issues were raised concerning the reconstructions and models. We therefore decided to already submit a revised manuscript with the old validation reports (which other than the recommended contour levels remain entirely valid), while we are updating the entries at the PDB.

REVIEWERS' COMMENTS

Reviewer #1 (Remarks to the Author):

Authors addressed all of my critiques thus this reviewer recommends publishing this work.

Reviewer #2 (Remarks to the Author):

All my minor concerns were adequately addressed. The revised manuscript by Zuber et al provides a structural model for how RfaH initially interacts with paused RNAP, how RfaH and its RNA target undergo structural changes following the initial encounter interaction, and how the original pause becomes a long-lived pause via backtracking. Biochemical studies demonstrate how GreA and Mfd are capable of reversing the backtracked complex into an elongation competent form. In my opinion this work is thorough, expertly executed, and will be of high interest to those interested in transcription.

Reviewer #3 (Remarks to the Author):

I feel that the authors have adequately addressed the reviewers' concerns and modify the manuscript accordingly. I have two minor recommendations.

1. I agree with the reviewer 2 that yellow is not an appropriate color, in particular when the figure is reproduced at low resolution. I suggest replacing with a color such as orange.
2. Can the authors add units to "time step of 0.002 and a temperature of 0.67" in line 820, page 33?

Reviewer #4 (Remarks to the Author):

See attachment.

In the manuscript “Concerted transformation of a hyper-paused transcription complex and its reinforcing protein” by Zuber et al., the authors performed structural investigations of the mechanism of transcription pausing with Cryo-EM structures. They leveraged the CryoEM structures before and after the RfaH refolding to investigate the fold-switch upon opsPEC binding. The structures and analysis presented are important for understanding the role of RfaH in transcription/translation. Regarding the MD simulations, the use of a dual-basin structure-based model is appropriate, given the large timescales for refolding in unbiased simulations. Still, a few minor changes to the statements are needed.

- 1) “Finally, our molecular dynamics simulations suggest that, following the initial domain separation, the RNAP-RfaH contacts are dispensable for the KOW α -to- β fold-switch.”. Since a biased Go-like model was used, some of the details regarding the RNAP contacts during refolding are lost or not needed. Since the expression of the KOW domain in solution leads to β fold structure similar to that in NusG, it is known that such a folding is spontaneous after domain separation. Therefore, the observation in MD may be argued as consistent, but probably not suggestive. The interactions during the domain separation are more interesting in this regard.
- 2) Given the availability of the full structures of the complexes, MD analysis could go beyond just focusing on refolding post-separation (i.e. focus of Fig.5), which has been extensively studied in the past. It’s not clear if the ops element contacts were included in the list of native contacts. Since this is considered to be the most important for triggering the domain separation, some description would be better.
- 3) Ref. 47 does not seem to use a dual-basin SBM as cited. Other relevant examples should be cited instead.
- 4) The opsPEC^{Rec} structure was used for both opsPEC-RfaH complexes, citing a larger number of missing residues. Couldn’t a model with an opsPEC^{Enc} template still be better? Perhaps comment for future investigations?
- 5) Please state the length of each trajectory (from Fig. 5, it seems $t=4,000$), either in the manuscript or Zenodo. What is its relevance in the actual (biological) timescale?

Response to Reviewer Comments

Reviewer comments are repeated in regular font, responses are in red, changed text passages are highlighted in yellow. Line numbers in responses refer to line numbers of the revised manuscript.

Reviewer #3 (Remarks to the Author):

2. Can the authors add units to “time step of 0.002 and a temperature of 0.67” in line 820, page 33?

We have added a description in the manuscript to indicate that our simulations are utilizing reduced units, as follows (line 832):

The dual-basin SBM simulations are run in reduced units, where the length scale, time scale, mass scale, and energy scale are all 1^{87} . Simulations were run using a time step of 0.002 reduced units (τ) and a temperature of 0.67 reduced units, collecting data every 1,000 timesteps (2τ) and with each simulation continued until the RMSD of RfaH against the cryoEM structure of the autoinhibited state reached values below 5 Å (maximum trajectory length = 11,600 τ ; Gaussian distribution mean trajectory length = 2,200 τ). Based on the comparison of SBMs with explicit solvent simulations, it is estimated that 1 τ is equivalent to 1 ns.

Reviewer #4 (Remarks to the Author):

In the manuscript “Concerted transformation of a hyper-paused transcription complex and its reinforcing protein” by Zuber et al., the authors performed structural investigations of the mechanism of transcription pausing with Cryo-EM structures. They leveraged the CryoEM structures before and after the RfaH refolding to investigate the fold-switch upon opsPEC binding. The structures and analysis presented are important for understanding the role of RfaH in transcription/translation. Regarding the MD simulations, the use of a dual-basin structure-based model is appropriate, given the large timescales for refolding in unbiased simulations. Still, a few minor changes to the statements are needed.

1) “Finally, our molecular dynamics simulations suggest that, following the initial domain separation, the RNAP-RfaH contacts are dispensable for the KOW α -to- β fold-switch.”. Since a biased Go-like model was used, some of the details regarding the RNAP contacts during refolding are lost or not needed. Since the expression of the KOW domain in solution leads to β fold structure similar to that in NusG, it is known that such a folding is spontaneous after domain separation. Therefore, the observation in MD may be argued as consistent, but probably not suggestive. The interactions during the domain separation are more interesting in this regard.

We rephrased this sentence as (line 96):

Finally, our molecular dynamics simulations are consistent with previous evidence that the KOW α -to- β fold-switch spontaneously occurs after the initial domain separation, with the RNAP-RfaH contacts being dispensable for the fold-switch.

2) Given the availability of the full structures of the complexes, MD analysis could go beyond just focusing on refolding post-separation (i.e. focus of Fig.5), which has been extensively studied in the past. It's not clear if the ops element contacts were included in the list of native contacts. Since this is considered to be the most important for triggering the domain separation, some description would be better.

The ops element contacts were indeed included, but when we analyzed the number of contacts formed against the DNA for RfaH, we noticed that they are all formed throughout the simulations, regardless of the folding state of RfaH, as show below.

In this regard, and alluding to the question #4, it would be useful to continue this work using both the opsPEC^{Enc} and opsPEC^{Rec} complexes for the simulations, which would enable to explore the encounter between RfaH and opsPEC prior to the refolding.

3) Ref. 47 does not seem to use a dual-basin SBM as cited. Other relevant examples should be cited instead.

Thanks for the comment. We decided to only keep the example of influenza hemagglutinin, because it utilizes a similar approach to our simulations.

4) The opsPEC^{Rec} structure was used for both opsPEC-RfaH complexes, citing a larger number of missing residues. Couldn't a model with an opsPEC^{Enc} template still be better? Perhaps comment for future investigations?

As indicated before, the availability of the opsPEC^{Enc} template would enable us to explore the binding of RfaH to ops and the subsequent triggering of the structural transformation of RfaH. We have added the following sentence in the Discussion (line 468):

Further work is required to fully explore the steps of NGN accommodation into its binding site and subsequent fold-switching.

5) Please state the length of each trajectory (from Fig. 5, it seems t=4,000), either in the manuscript or Zenodo. What is its relevance in the actual (biological) timescale?

As stated in lines 828-832, each simulation continued until the RMSD of RfaH against the cryoEM structure of the autoinhibited state reached values below 5 Å. We have now added the Gaussian distribution mean trajectory length and maximum trajectory length of the trajectories.

Regarding the relevance in the actual biological timescale, this is difficult to address due to the use of (1) reduced units, and (2) a dual-basin approach that effectively bias the system towards the active state of RfaH. While we cannot suggest a biological timescale for the refolding processes explored herein, we do cite a work in which it is estimated that 1 reduced unit is equivalent to 1 ns.